# Artificially decreasing cortical tension generates aneuploidy in mouse oocytes

Isma Bennabi [1], Flora Crozet[1,7], Elvira Nikalayevich[1,7], Agathe Chaigne [2], Gaëlle Letort [1], Marion Manil-Ségalen[1], Clément Campillo [3], Clotilde Cadart [4,5], Alice Othmani[6], Rafaele Attia[4,5], Auguste Genovesio [6], Marie-Hélène Verlhac [1]✉ & Marie-Emilie Terret [1]✉

Human and mouse oocytes' developmental potential can be predicted by their mechanical properties. Their development into blastocysts requires a specific stiffness window. In this study, we combine live-cell and computational imaging, laser ablation, and biophysical measurements to investigate how deregulation of cortex tension in the oocyte contributes to early developmental failure. We focus on extra-soft cells, the most common defect in a natural population. Using two independent tools to artificially decrease cortical tension, we show that chromosome alignment is impaired in extra-soft mouse oocytes, despite normal spindle morphogenesis and dynamics, inducing aneuploidy. The main cause is a cytoplasmic increase in myosin-II activity that could sterically hinder chromosome capture. We describe here an original mode of generation of aneuploidies that could be very common in oocytes and could contribute to the high aneuploidy rate observed during female meiosis, a leading cause of infertility and congenital disorders.

[1] CIRB, Collège de France, UMR7241/U1050, 75005 Paris, France. [2] MRC Laboratory for Molecular Cell Biology, UCL, London WC1E 6BT, UK. [3] LAMBE, Université d'Evry val d'Essonne, UMR 8587 Evry, France. [4] Institut Curie, PSL Research University, CNRS, UMR 144, F-75005 Paris, France. [5] Institut Pierre-Gilles de Gennes, PSL Research University, F-75005 Paris, France. [6] IBENS, Ecole Normale Supérieure, UMR8197/U1024, 75005 Paris, France. [7]These authors contributed equally: Flora Crozet, Elvira Nikalayevich. ✉email: marie-helene.verlhac@college-de-france.fr; marie-emilie.terret@college-de-france.fr

Meiosis in human females is error prone, resulting in aneuploidy, and as such is the leading cause of miscarriage and developmental disabilities such as trisomies[1]. It was recently shown that human and mouse oocytes developmental potential is accurately predicted by mechanical properties within hours after fertilization[2,3]. In previous studies, we have deciphered how cortex tension is regulated in mouse oocytes and embryos[4–6]. Using multidisciplinary approaches, we showed that the nucleation of a cortical F-actin thickening by the Arp2/3 complex in meiosis I excludes myosin-II from the cortex, decreasing cortical tension[4,5]. This change in cortex mechanics amplifies an initial imbalance of potential pulling forces exerted by myosin-II at the actin cage surrounding the microtubule (MT) spindle[5,7–9], the forces being probably stronger at the pole closest to the cortex because of the initial slight asymmetry of spindle position[10]. Spindle motion is slow[8,10] but is amplified by the progressive deformation of the cortex, made possible by lowering of cortical tension. This reduced cortex tension allows the recruitment of filaments between the cortex and the spindle, and therefore potential amplification of initial forces. The change in cortex mechanics is required for spindle migration from the oocyte center to its cortex, generating an asymmetric division in size after anaphase. The asymmetry in size of the meiotic division allows the preservation of maternal stores accumulated during oocyte growth and required for embryo development. Although the drop in cortex tension is required for spindle migration in oocytes[4], spindle migration is also prevented by a too low tension[5]. Thus, the geometry of the division depends on a narrow window of cortical tension, regulated by myosin-II cortical localization, itself fine-tuned by actin nucleation. Defects in cortical tension affect the geometry of oocyte divisions, potentially impacting on their developmental potential[11]. In this study, we address the origin of early developmental failure due to cortical tension defects in the oocyte other than perturbation in the geometry of the division. We focused on extra-soft oocytes, which represent the most common case in a natural population of human and mouse oocytes[2]. To specifically enrich in this subpopulation to be able to analyze a large number of cells, we generated extra-soft oocytes by using two constructs promoting a decrease in cortical tension when expressed in mouse oocytes. First, we used the cortical verprolin-homology cofilin-homology acidic (cVCA) construct[5] that forces Arp2/3-dependent actin nucleation at the cortex of oocytes, chasing myosin-II from the cortex, leading to cortical tension decrease and apparent softening[4,5]. Second, we designed a construct that induces ectopic actin assembly by the FH1FH2 nucleating domain of formin 2 at the cortex of oocytes. We show that expression of this construct also induces myosin-II cortical displacement and reduces cortical tension.

Using these two approaches, we observe impaired chromosome alignment and aneuploidy in extra-soft oocytes, despite normal spindle assembly and dynamics as assessed by MT density and growth measurements. By performing laser ablation experiments, we show the presence of forces applied by F-actin on the spindle and transmitted to the chromosomes. Even though the intensity of these forces appears reduced in extra-soft oocytes, they do not seem to contribute to chromosome misalignment. Rather, a cytoplasmic increase in myosin-II activity appears as the main cause of chromosome misalignment, potentially sterically hindering chromosome capture, as decreasing myosin-II activity in extra-soft oocytes rescues alignment. In this work, we describe a possible mode of generation of aneuploidies that could be very common in female gametes. Indeed, 36% of oocytes are measured as too soft in a natural population[2]. A fraction of these naturally soft oocytes might present chromosome alignment defects impeding on their future development after fertilization,

contributing to the high aneuploidy rate measured in female meiosis, a leading cause of infertility[1].

## Results

**Too soft oocytes are aneuploid.** Extra-soft oocytes are the most represented case in a natural population of mouse and human oocytes[2]. To specifically enrich in this subpopulation and be able to analyze a large number of cells, we developed tools able to decrease cortical tension.

The first one is the cVCA (Fig. 1a), a construct that forces branched Arp2/3-dependent nucleation of F-actin at the cortex, chasing cortical myosin-II and leading to a decrease in cortical tension[5]. As a consequence, cVCA-expressing oocytes (named cVCA oocytes hereafter) are extra soft. We followed chromosome alignment and segregation by live spinning-disk microscopy during meiosis I in cVCA oocytes. Whereas chromosomes are aligned on the metaphase I plate in controls, cVCA oocytes display chromosome alignment defects before anaphase I (Fig. 1b). We defined oocytes with misaligned chromosomes before anaphase I as having one or several chromosomes away from the metaphase plate. Using this criterion, 48% of cVCA oocytes present chromosomes that are not aligned before anaphase I (Fig. 1c right gray bar) compared with 15 % in controls (Fig. 1c left gray bar). cVCA oocytes still undergo anaphase on time, despite having misaligned chromosomes[5]. This leads to aberrant chromosome segregation at anaphase I (Fig. 1b red asterisk) and aneuploidy (Supplementary Fig. 1 80% of oocytes are euploid in controls vs. 37.5% in cVCA oocytes and Supplementary Movie 1), as measured on intact oocytes using Monastrol spreads[12] (see Methods). To further characterize the phenotype of cVCA oocytes, we developed an automated and blind approach to discriminate statistically significant features different between control and extra-soft oocytes. This was achieved using an unbiased computational imaging approach to automatically threshold the stacks of images (control and cVCA oocytes 30 min before anaphase I) and extract the features differing the most between controls and extra-soft oocytes (as in ref. [13]). The aspect ratio was the most significantly different feature between the two conditions (Fig. 1d). This feature is defined by the ratio of the minor axis length to the major axis length. When this ratio is 1, both axes are equal and the shape is close to a circle (Fig. 1d red circle on the right). On the opposite, when the ratio is close to 0, the minor axis is much smaller than the major axis and the shape is flat (Fig. 1d red ellipse on the left). For controls, the aspect ratio was close to 0 (Fig. 1d black curve bars), showing that the chromosomes were aligned. In cVCA oocytes, aspect ratios formed a bimodal distribution between 0 and 1, arguing that some of the oocytes displayed chromosomes that were aligned as in the controls but others displayed chromosomes that were misaligned (Fig. 1d blue curve bars). This computational approach allowed us to quantify and validate the chromosome alignment defect we observed in cVCA oocytes, and allowed us to find an easy criterion to measure it in an unbiased manner (Fig. 1d). This criterion was then applied manually in the rest of the study by drawing the bounding boxes around the chromosomes and measuring their width as a readout for chromosome alignment (Fig. 1h).

To exclude the possibility that impaired chromosome alignment was specific to cVCA-expressing oocytes, we designed another tool able to decrease cortical tension, the cFH1FH2. This construct was obtained by fusing the FH1FH2 domain of formin 2 to ezrin (Fig. 1e), forcing linear actin nucleation at the cortex of mouse oocytes. Similar to the cVCA, the cFH1FH2 localizes specifically to the oocyte cortex (Supplementary Fig. 2A bottom left panel). Unlike the cVCA and probably due to the difference

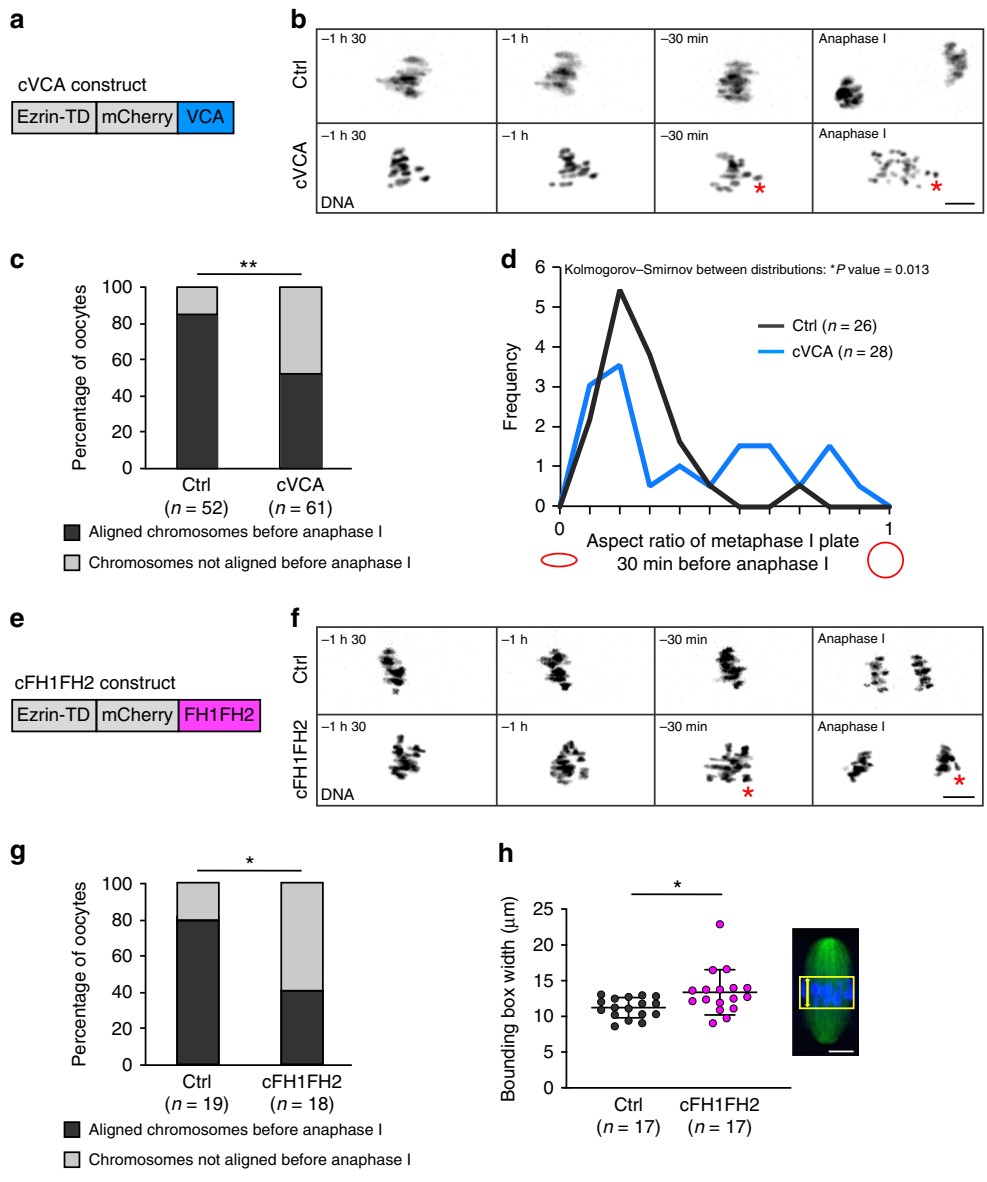

**Fig. 1 Extra-soft oocytes have chromosome alignment defects in metaphase I. a** Scheme of the cVCA construct. **b** Representative time-lapse movies of chromosomes (black, (H2B)-GFP) in control and cVCA oocytes starting 1 h 30 min before anaphase I (10 independent experiments). Scale bar: 10 μm. The red asterisk points at a misaligned chromosome segregated in the oocyte. **c** Bar graph representing the percentage of oocytes with aligned (black) or not (gray) chromosomes 30 min before anaphase I for controls and cVCA oocytes. $n$ is the number of oocytes analyzed. Data are from ten independent experiments. Statistical significance of differences is assessed with a two-sided $\chi^2$-test: **$P$-value = 0.0003. **d** Graph representing the aspect ratio (close to 1 is a circle, close to 0 an ellipse as shown in red) for controls and cVCA oocytes. Quantifications 30 min before anaphase I. $n$ is the numbers of oocytes analyzed. Data are from seven independent experiments. Statistical significance of differences between distributions is assessed with a two-sided Kolmogorov–Smirnov test: *$P$-value = 0.013. **e** Scheme of the cFH1FH2 construct. **f** Representative time-lapse movies of chromosomes (black, (H2B)-GFP) in control and cFH1FH2 oocytes starting 1 h 30 min before anaphase I (three independent experiments). Scale bar: 10 μm. The red asterisk points at a misaligned chromosome segregated in the oocyte. **g** Bar graph representing the percentage of oocytes with aligned (black) or not (gray) chromosomes 30 min before anaphase I for controls and cFH1FH2 oocytes. $n$ is the number of oocytes analyzed. Data are from three independent experiments. Statistical significance of differences is assessed with a two-sided Fisher's test: *$P$-value = 0.0201. **h** Dot plot showing the width of the bounding box containing the chromosomes 30 min before anaphase I in controls and cFH1FH2 oocytes. $n$ is the number of oocytes analyzed. Data obtained from three independent experiments. Black bars and whiskers represent mean and SD. Statistical significance of differences is tested with a two-sided Mann–Whitney test: *$P$-value = 0.0134. The bounding box corresponds to the yellow square in the image of spindle (green) with chromosomes (blue), its width to the yellow dashed line. Scale bar: 5 μm.

in the nature of the network nucleated (branched for cVCA vs. linear for cFH1FH2), oocytes expressing the cFH1FH2 (named cFH1FH2 oocytes hereafter) do not nucleate a cortical actin thickening (Supplementary Fig. 2A bottom right panel and quantification in 2B). Surprisingly, expression of the cFH1FH2 construct is associated with a reduction in cortical myosin-II

levels (Supplementary Fig. 2C and quantification in 2D). These data show that overnucleation of linear actin by FH1FH2 at the cortex of oocytes is sufficient to chase myosin-II. It suggests that myosin-II can be chased by steric hindrance and not by preferential binding to one type of actin network (linear vs. branched) at the cortex of mouse oocytes. Finally, cFH1FH2

expression induces an eight-fold decrease in cortical tension measured by micropipette aspiration (Supplementary Fig. 2E, $3.58 \pm 1.1$ nN per µm for controls and $0.41 \pm 0.2$ nN per µm for cFH1FH2 oocytes). Thus, cortical myosin-II removal in cFH1FH2 oocytes leads to cortex softening as for cVCA oocytes. Similar to what was observed with the cVCA construct, meiosis I spindle length is comparable to controls in cFH1FH2 oocytes (Supplementary Fig. 2F), but its migration is impaired (Supplementary Fig. 2G and quantification in 2H), consistent with the fact that cortical tension impacts meiosis I spindle migration in mouse oocytes[4,5]. Very interestingly, cFH1FH2-expressing oocytes also display chromosome alignment defects (Fig. 1f). In controls, chromosomes are aligned on the metaphase plate at the end of meiosis I (Fig. 1f upper panels), whereas chromosomes in cFH1FH2 oocytes do not form a proper plate (Fig. 1f bottom panels). In particular, 61% of cFH1FH2 oocytes have misaligned chromosome before anaphase against 21% of controls (Fig. 1g). To further quantify chromosome alignment, we measured the width of the bounding box encompassing the metaphase I plate (Fig. 1h yellow square and yellow dash line). If chromosomes are misaligned, the bounding box width should be larger than when chromosomes are tightly aligned. Indeed, the bounding box width is increased before anaphase in cFH1FH2 oocytes ($13.37 \pm 3.1$ µm; Fig. 1h magenta dots) compared with controls ($11.23 \pm 1.4$ µm; Fig. 1h black dots). cFH1FH2 oocytes still undergo anaphase and first polar body extrusion (Fig. 1f and Supplementary Fig. 2I), despite having misaligned chromosomes, leading to aberrant chromosome segregation at anaphase I (Fig. 1f red asterisk).

In conclusion, actin accumulation at the oocyte cortex, regardless of its organization, sterically excludes myosin-II, decreasing cortical tension. Remarkably, chromosome alignment errors could be a hallmark of oocytes with a too low cortical tension.

**Too soft oocytes, smaller, have normal cytoplasmic activity.** As observed before[5], cVCA oocytes display strong shape deformations inherent to their decrease in cortical tension (Supplementary Fig. 3A). The same is true for cFH1FH2 oocytes (Supplementary Fig. 2A, C). These deformations are stable at long time scales (Supplementary Fig. 3B spanning 1 h 30 min) and at short time scales, as observed by high-frequency video microscopy (Supplementary Movie 2). The presence of deformations in cVCA oocytes prompted us to precisely measure their volume. For this, we took advantage of the fluorescence exclusion measurement (FXm) method initially described in ref. [14], to measure the volume of control and cVCA oocytes (Supplementary Fig. 3C). The mean volume of cVCA oocytes is reduced by 9.45% compared with controls (Supplementary Fig. 3D).

We then wondered what could be the consequences of this volume reduction of about 10%, and in particular if it could impact cytoplasmic activity. Cytoplasmic activity represents the movement of the cytoplasm within a cell. The cytoplasmic motion translates into the movement of all intracellular organelles, as has been observed in Prophase I mouse oocytes[15]. One could imagine that extra-soft oocytes with a reduced volume would have a more viscous cytoplasm, reflected by a lower cytoplasmic activity, impairing chromosome movement and resulting in chromosome misalignment. To test this hypothesis, we assessed oocyte cytoplasmic activity by tracking the movement of vesicles using high-frequency spinning-disk microscopy (Supplementary Movie 3), as previously described in ref. [15]. We measured the mean square displacement (MSD) of the vesicles, representing the space they explore, obtained by averaging the square of the distance traveled per unit of time. The MSD plots describing vesicle movement in the cytoplasm are similar,

suggesting that the cytoplasmic activity is not impaired in cVCA oocytes (Supplementary Fig. 3E). In conclusion, chromosome misalignment in cVCA oocytes is not due to a difference in cytoplasmic activity.

**Too soft oocytes transmit weaker forces to chromosomes.** Defects in chromosome alignment could be due to altered forces transmitted from the cortex to the chromosomes. We first directly tested whether forces are mechanically transmitted to chromosomes in control oocytes and then whether a decrease in cortical tension results in aberrant forces, potentially impairing chromosome alignment in extra-soft oocytes.

In control oocytes, chromosomes migrate from the oocyte center to its cortex. This migration depends only on F-actin[7,8,10,16,17] and not on the presence of astral MTs as in mitotic cells[18–20], as oocytes are devoid of centrioles and astral MTs[21,22]. Actin is organized in a cytoplasmic network including an actin cage around the MT spindle, which is connected to the subcortical actin network[5,7–9] (Supplementary Fig. 4A). The force generator element for spindle migration is likely myosin-II, which seems to exert pulling forces on the actin cage surrounding the MT spindle as shown using inhibitors[5,8]. However, there is no direct evidence that forces are applied on the spindle by actomyosin networks in oocytes. We assessed qualitatively these forces by performing laser ablation. The F-actin cytoplasmic network was cut between the spindle pole and the closest cortex at the end of meiosis I, 6 h 30 min after nuclear envelope breakdown (BD + 6 h 30 min) on several Z stacks spanning the entire thickness of the metaphase plate (Supplementary Fig. 4A yellow dotted line, Supplementary Fig. 4B, C, and Supplementary Movies 4 and 5). Laser ablation did not damage oocytes, as they all extruded a polar body and arrested in metaphase II normally following ablation. After laser ablation, the response of Histone (H2B)-GFP labeled chromosomes was monitored by live microscopy during 1 min. MTs were visualized using very low doses of EB3-GFP to follow the spindle and make sure that the ablation was not performed within the spindle but away from spindle poles (Supplementary Fig. 4B, C and Supplementary Movies 4 and 5). Before laser ablation, the chromosomes are clustered and aligned on the metaphase plate in control oocytes at BD + 6 h 30 min (Supplementary Fig. 4B upper left panel $t = 0$ min). One minute after laser cutting, the metaphase plate shrinks (Supplementary Fig. 4B upper middle panel). To quantify this effect, we measured the width of the bounding box encompassing the metaphase plate before and after laser ablation (Supplementary Fig. 4B in plain and dotted yellow lines, respectively). The width of the bounding box decreases progressively after ablation (Supplementary Fig. 4B upper panels and Supplementary Fig. 4D black curve) arguing that the actomyosin networks exerts forces on the chromosomes. In contrast, when we performed laser ablation at BD + 3 h before spindle-cortex attachment[7,23], we did not see an impact on chromosome behavior (Supplementary Fig. 4E and Supplementary Movie 6). In addition, we never observed progressive decrease in the width of the bounding box without laser ablation in oocytes at BD + 6 h 30 min (Supplementary Fig. 4F and Supplementary Movie 7). Our results demonstrate that the actomyosin network exerts forces on the chromosomes, which are indeed responsible for the motion of the spindle to the cortex in mouse oocyte.

Previous work showed that spindle migration is impaired in extra-soft oocytes, suggesting that the forces applied on the spindle are altered[5]. This was also predicted by a theoretical model, but never directly tested experimentally[5]. In cVCA oocytes, chromosomes are scattered on the metaphase plate before ablation at BD + 6 h 30 min (Supplementary Fig. 4B

bottom left panel $t = 0$ min). After cutting of the spindle-cortex connection, the chromosomes remain scattered on the metaphase plate, which stays stable in width (Supplementary Fig. 4B bottom panels, Supplementary Fig. 4D blue curve, and Supplementary Movie 4). This suggests that, at that stage, the intensity of the forces applied on the chromosomes is reduced in cVCA oocytes compared with controls. Our results validate the mathematical modeling of spindle migration predicting that the intensity of forces transmitted to the spindle are diminished in extra-soft oocytes[5]. We then wondered whether the decrease in forces applied to the meiotic spindle could have consequences on chromosome alignment.

To test this, we turned to another type of oocytes, the $Fmn2^{-/-}$ oocytes, which have reduced forces applied on their spindle. Indeed, $Fmn2^{-/-}$ oocytes are invalidated for the cytoplasmic actin nucleator formin 2 and lack the cytoplasmic network and the actin cage[7,8]. As a consequence, the spindle is not attached to the cortex and does not migrate[16,24]. Thus, in $Fmn2^{-/-}$ oocytes, the actomyosin networks do not apply forces on the spindle or chromosomes. However, $Fmn2^{-/-}$ oocytes do not have chromosome alignment defects in metaphase I[9] (Supplementary Fig. 4G). In conclusion, it is unlikely that the reduction in the intensity of the forces applied to chromosomes in cVCA oocytes is the main mechanism involved in the generation of misaligned chromosomes.

**Spindle morphogenesis is normal in extra-soft oocytes.** To gain insight into the origin of the chromosome alignment defects, we observed how spindle morphogenesis proceeded in extra-soft oocytes. Indeed, in mitotic cells, aberrant cortical tension impairs spindle formation, leading to chromosome segregation errors[25]. We monitored spindle morphogenesis by live spinning-disk microscopy during meiosis I in mouse oocytes incubated with SiR-Tubulin to label MTs (Fig. 2a and Supplementary Movie 8). Although they have misaligned chromosomes (Fig. 2a bottom panels), the spindle appears to form properly in cVCA oocytes as in controls (Fig. 2a top panels).

We then measured relative MT densities and MT growth early on during spindle morphogenesis, since defects at this stage are known to induce chromosome misalignment later on[26–28]. Control and cVCA oocytes expressing EB3-GFP, a MT plus-end tracker, were imaged at $BD + 2$ h after Monastrol treatment to inhibit spindle bipolarization (as in ref. [29], Supplementary Movie 9). MT densities and monoaster sizes are comparable in control (gray dots) and extra-soft (blue dots) oocytes (Fig. 2b dot plots and yellow dashed circles). Tracking of individual MT plus ends (Fig. 2c red tracks) shows that MT growth rates are also comparable in control (gray dots) and extra-soft oocytes (blue dots in Fig. 2c). Consistent with these observations, the spindle length of cVCA oocytes is comparable to controls at $BD + 7$ h ($29.93 \pm 1.64$ μm in cVCA oocytes compared with $29.27 \pm 1.80$ μm in controls; Fig. 2d).

All these data indicate that spindle morphogenesis and dynamics are comparable to controls in extra-soft cVCA oocytes. Moreover, it was shown previously that cVCA expression does not alter the architecture nor the dynamics of the cytoplasmic actin network[5], two features that could potentially affect spindle formation when they are modified[23].

**Chromosome capture is less efficient in extra-soft oocytes.** As we did not observe gross spindle defects, to further explore the reason for chromosome alignment defects in extra-soft oocytes, we performed a more thorough analysis of chromosome movements. For this, we tracked individual chromosomes on the metaphase I plate during 20 min at $BD + 6$ h 30 min. Individual

chromosome tracks suggest that chromosomes explore more space in metaphase I cVCA oocytes than in controls (Fig. 3a for representative examples and Supplementary Movie 10), which is confirmed by their MSD analysis (Fig. 3b). Fitting of the MSD slopes to a simple linear model and statistical comparison of the slopes revealed different diffusion coefficient in control vs. cVCA oocytes (Fig. 3b) demonstrating that chromosomes in cVCA oocytes explore more space than in controls. This could reflect a problem in chromosome capture.

To test this hypothesis, we analyzed chromosome distribution throughout meiosis I in control and cVCA oocytes from NEBD to $BD + 7$ h 30 min. To do so, we fitted an ellipse around the chromosomes (Supplementary Movie 11), measured the length of its major axis and compared its variation over a window of 2 h (Fig. 3c). This variation is a measure of chromosome alignment efficiency: if it is negative, it means that the ellipse shortens reflecting chromosome gathering. In control and cVCA oocytes, the variation is positive between NEBD and $BD + 2$ h, indicating that the ellipse elongates, corresponding to chromosome condensation and individualization (as described in ref. [30]). Starting between $BD + 2$ h and 2 h 30 min, the variation becomes negative in the controls (black curve), showing that the ellipse shortens, suggesting that chromosomes are converging/aligning. In cVCA oocytes, the variation is close to 0 and significantly different from the controls at $BD + 3$ h and $BD + 3$ h 30 min (blue curve), showing that the ellipse does not vary, suggesting that chromosome alignment is less efficient or slower. The phase between $BD + 2$ h 30 min and 4 h corresponds to chromosome capture and the first chromosome biorientation attempts[30], suggesting that chromosomes are less efficiently captured in cVCA oocytes despite normal MT density and growth (Fig. 2b, c).

If it is the case, chromosomes should experience less tension across opposite kinetochores of bivalents in cVCA oocytes. To test that, we measured the distance between major satellite repeats in each bivalent at $BD + 6$ h 30 min (Fig. 3d and Supplementary Movie 12), an indirect measure of tension across bivalents[31]. We represent a distribution graph showing the percentage of bivalents falling in a certain range of major satellite repeats distances (Fig. 3e). For cVCA oocytes, the curve (Fig. 3e blue) is shifted towards smaller distances, showing that cVCA bivalents experience less tension, consistent with the fact that chromosomes are less efficiently captured in cVCA oocytes.

**Accumulation of myosin-II generates chromosome misalignment.** It was shown recently in starfish oocytes that actin structures can sterically hinder kinetochore-MT attachments, a physiological process important to coordinate chromosome capture by MTs[32]. We thus wondered whether such a mechanism could be at play in extra-soft oocytes, leading to less efficient chromosome capture. It was shown previously that the expression of the cVCA does not alter the architecture nor the dynamics of the cytoplasmic actin network[5]. However, we analyzed in more detail actin organization in extra-soft oocytes at later stages and tested whether depolymerizing actin could rescue chromosome alignment. We visualized F-actin at $BD + 6$ h 30 min by staining oocytes with fluorescent phalloidin (Fig. 4a gray). F-actin forms a cytoplasmic network and an actin cage in controls, as previously described[7,8] (Fig. 4a left panels). The same organization can be observed in cVCA oocytes (Fig. 4a right panels). We quantified F-actin fluorescent intensity levels on the spindle and in the cytoplasm in control and cVCA oocytes (Fig. 4b, c). We could not detect any significant difference between control and cVCA oocytes, suggesting that chromosome capture in cVCA oocytes is not hindered by an accumulation of F-actin. Accordingly,

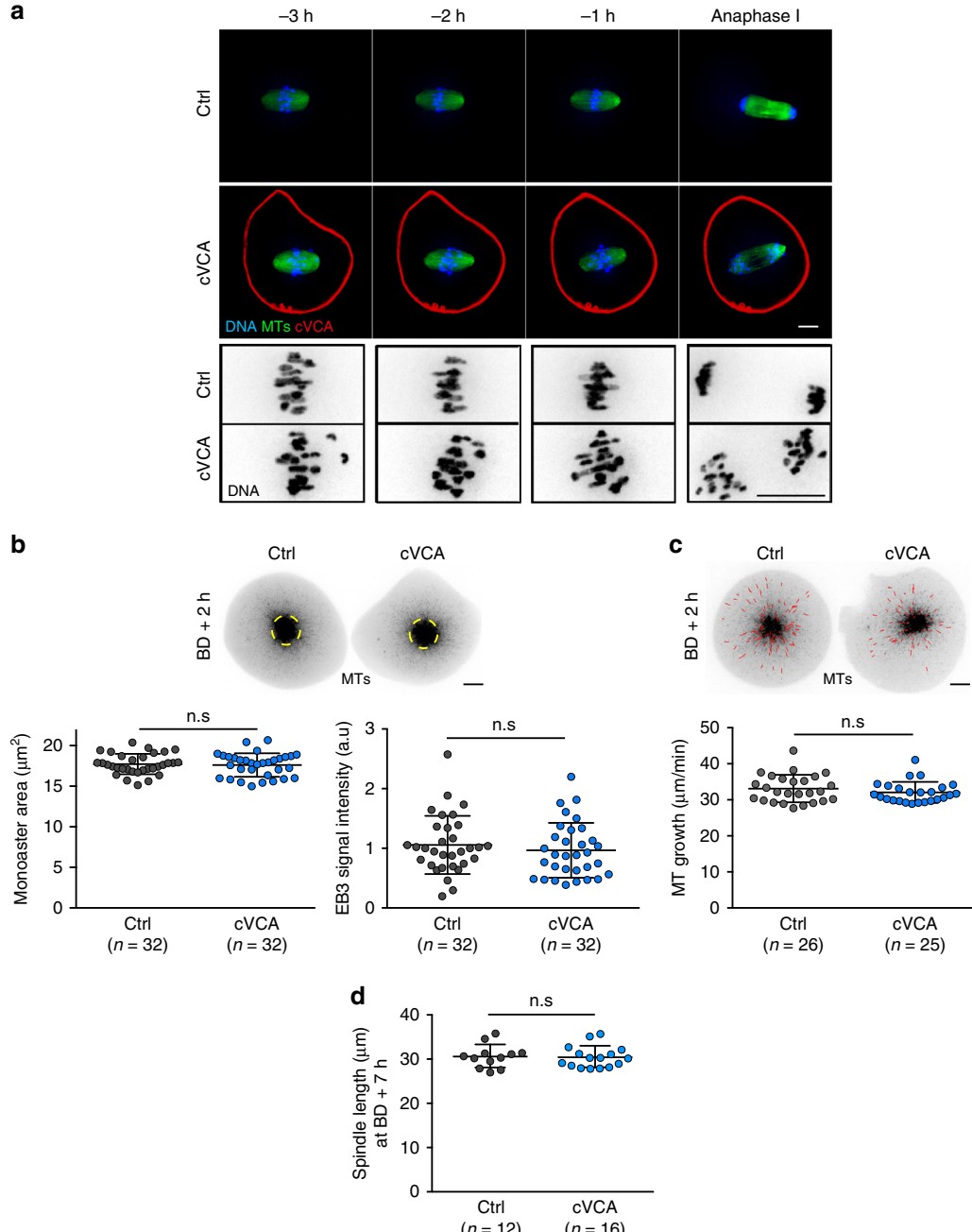

**Fig. 2 Spindle morphogenesis is normal in extra-soft oocytes. a** Representative time-lapse movies of control and cVCA (red) oocytes expressing Histone (H2B)-GFP (blue). Microtubules (MTs, green) are visualized with SiR-Tubulin. Movies start 3 h before anaphase I. The bottom rows show magnification of the chromosomes (black) from the upper panels. Scale bars: 10 μm. Four independent experiments. **b** Top panel: control and cVCA oocytes observed at BD + 2 h expressing EB3-GFP (black) and treated with Monastrol. Scale bar: 10 μm. Bottom panels: dot plots showing the area (left panel) and fluorescence intensity (right panel) of the EB3-GFP monoasters (yellow dashed circles highlighted on the top panels) in controls and cVCA oocytes. Data are from three independent experiments. Statistical significance is determined with an unpaired *t*-test for monoaster size n.s *P*-value = 0.7336 and a two-sided Mann–Whitney test for monoaster fluorescence intensity n.s *P*-value = 0.3871. **c** Top panel: control and cVCA oocytes observed at BD + 2 h expressing EB3-GFP (black) treated with Monastrol. Scale bar: 10 μm. EB3-GFP comets were tracked on a single plane and average comet speed per oocyte is displayed on the dot plot (bottom panel), reflecting MT growth. Example of tracks in control and cVCA oocytes are highlighted in red (top panel). Data are from three independent experiments. Statistical significance is determined with a two-sided Mann–Whitney test: n.s *P*-value = 0.3739. **d** Spindle length quantifications for controls and cVCA oocytes 7 h after nuclear envelope breakdown (BD + 7 h). *n* is the number of oocytes analyzed. Data are from four independent experiments. Statistical significance is determined with a two-sided Mann–Whitney test: n.s. *P*-value = 0.432. For **b**, **c**, **d**, black bars and whiskers represent mean and SD.

depolymerizing F-actin in extra-soft oocytes does not rescue chromosome alignment defects (Fig. 4d, e), suggesting also that F-actin is not the cause of chromosome misalignment in extra-soft oocytes.

We thus wondered what could hinder chromosome capture in extra-soft oocytes and hypothesized that it could be an excess of myosin-II in the cytoplasm. Indeed, oocytes expressing the cVCA or the cFH1FH2 constructs share chromosome alignment defects

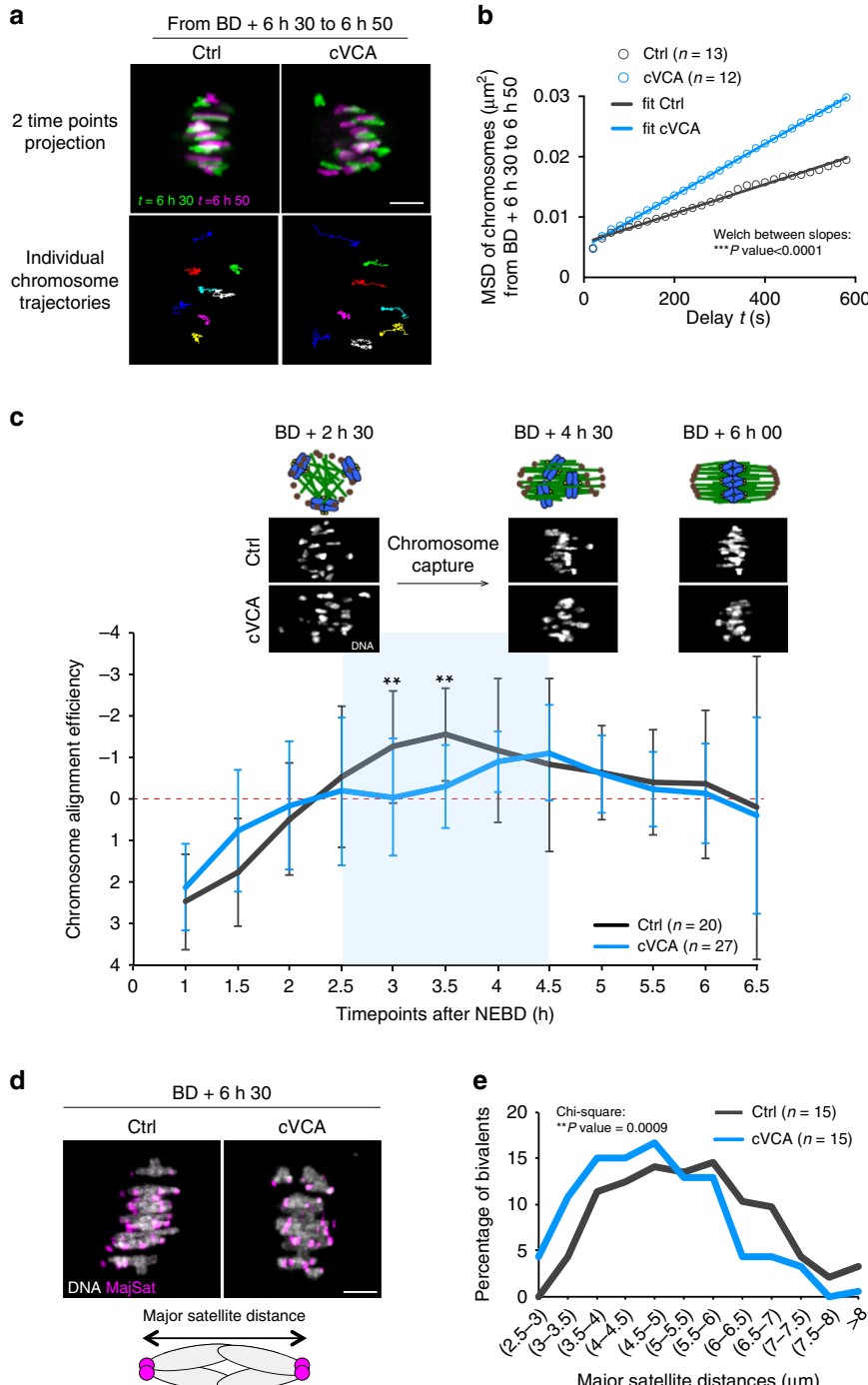

**Fig. 3 Chromosome capture is less efficient in extra-soft oocytes. a** Representative temporal color-coded projections from time-lapse movies of a control and cVCA oocyte (upper panels). Scale bar: 5 μm. Bottom panels show individual chromosome trajectories corresponding to the upper panels. Each color represents one chromosome track. Four independent experiments. **b** Graph representing the mean square displacement (MSD) of individual chromosome between BD + 6 h 30 min and 6 h 50 min. Fifty-nine chromosomes among 13 controls and 60 chromosomes among 12 cVCA oocytes were analyzed. Data are from four independent experiments. MSD data are fitted to a simple linear regression model ($R^2 > 0.97$). Statistical significance of differences of slopes between Ctrl and cVCA is assessed with a two-sided Welch's test: ***$P$-value < 0.0001. **c** Graph representing the variation of elongation of the ellipse fitting the chromosomes between nuclear envelope BD (NEBD) and BD + 7 h 30 min for controls and cVCA oocytes. $n$ is the number of oocytes analyzed. Data are from five independent experiments. The mean and SD are shown. Statistical significance of differences is assessed with a two-sided Wilcoxon's test: **$P$-value = 0.005. The blue part of the graph corresponds to the chromosome capture phase. Images (chromosomes in white) and corresponding schemes (microtubules in green, DNA in blue, microtubule organizing centers in brown) are shown for BD + 2 h 30 min, 4 h 30 min, and 6 h. **d** Representative projections from time-lapse movies of control and cVCA oocytes at BD + 6 h 30 min expressing Histone(H2B)-GFP (gray) and MajSat-mClover (magenta). Scale bar: 5 μm. The scheme represents a bivalent (major satellite repeats in magenta). Five independent experiments. **e** Distribution graph representing the distance between major satellite repeats in live oocytes at BD + 6 h 30 min for controls and cVCA oocytes. One hundred and eighty-five chromosomes among 15 controls and 186 chromosomes among 15 cVCA oocytes were analyzed. Data are from five independent experiments. Statistical significance of differences is assessed with a two-sided $\chi^2$-test. **$P$ = 0.0009.

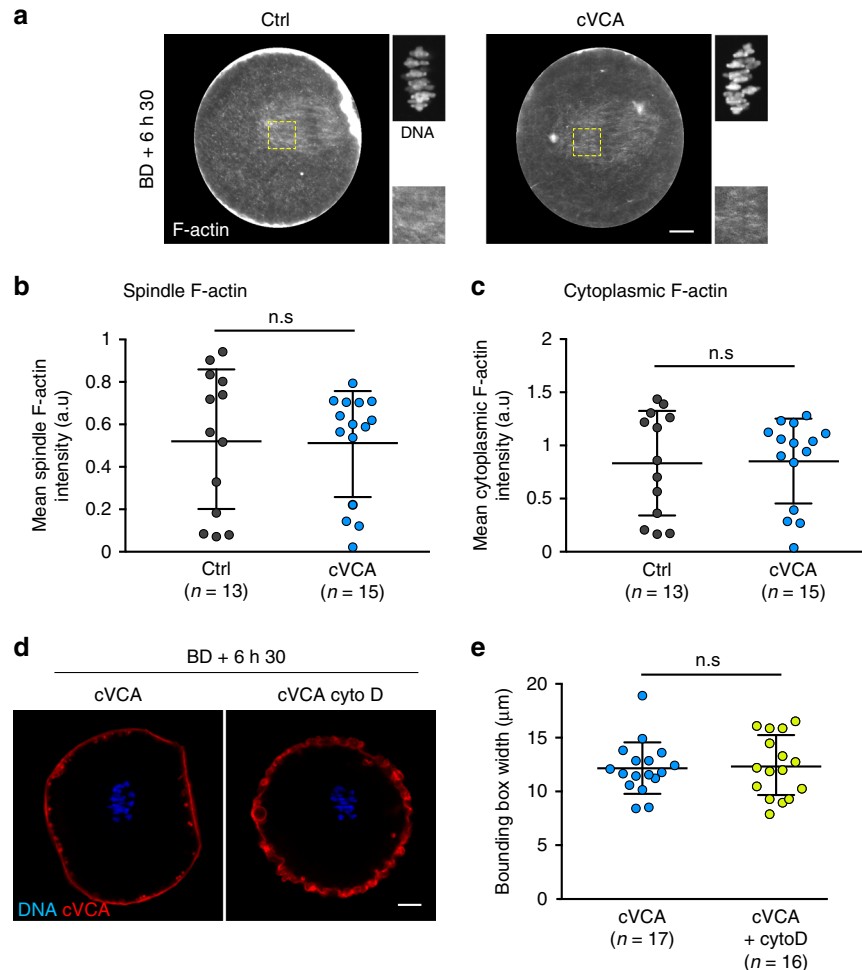

**Fig. 4 Actin organization and density in extra-soft oocytes is similar to controls and its depolymerization does not rescue chromosome alignment.**
**a** Cropped images of representative immunofluorescence of control and cVCA oocytes stained for F-actin (gray) at BD + 6 h 30 min. Insets are magnification of the chromosomes (gray) and of regions marked by yellow dashed line boxes. Scale bar: 10 µm. Three independent experiments. **b** Dot plot representing the F-actin spindle intensity at BD + 6 h 30 min in controls and cVCA oocytes. n is the number of oocytes analyzed. Data are from three independent experiments. Statistical significance of differences is assessed with a two-sided Mann–Whitney test: n.s P-value = 0.6832. **c** Dot plot representing the F-actin cytoplasm intensity at BD + 6 h 30 min in controls and cVCA oocytes. n is the number of oocytes analyzed. Data are from three independent experiments. Statistical significance of differences is assessed with a two-sided Mann–Whitney test: n.s P-value = 0.8207. **d** Oocytes at BD + 6 h 30 min expressing Histone(H2B)-GFP (blue) and cVCA (red). The oocyte on the right panel were treated with 1 µg/ml cytochalasin D. Scale bar: 10 µm. Five independent experiments. **e** Dot plot representing the metaphase I plate width at BD + 6 h 30 min in cVCA oocytes treated or not with 1 µg/ml cytochalasin D. n is the number of oocytes analyzed. Data are from five independent experiments. Statistical significance of differences is assessed with a two-sided Mann–Whitney test: n.s P-value = 0.810. For **b**, **c**, **e**, black bars and whiskers represent mean and SD.

(Fig. 1b, f) and chase precociously myosin-II from the cortex (Supplementary Figs. 2C, D and 5) leading to a softened cortex[5] (Supplementary Figure 2E). First, we quantified myosin-II accumulation over time in the cytoplasm in control and extra-soft oocytes (Supplementary Fig. 5). We followed endogenous myosin-II using a specific GFP-coupled intrabody directed against myosin-II[4,33] (Supplementary Fig. 5A) and measured the fluorescence signal intensity in the cytoplasm 2 h and 9 h after cRNA injection in control and extra-soft oocytes (Supplementary Fig. 5B). Levels of cytoplasmic myosin-II are comparable 2 h after cRNA injection in all conditions (control, cVCA, cFH1FH2, Supplementary Fig. 5B). Nine hours after cRNA injection, extra-soft oocytes have 1.9 (for cFH1FH2) and 2.5 (for cVCA oocytes) times more myosin-II in their cytoplasm compared with control oocytes at the same stage (Supplementary Fig. 5B). In addition, extra-soft oocytes accumulate more myosin-II in their cytoplasm over time compared with controls (1.5 times accumulation of myosin-II for controls, 2.1 for cFH1FH2, and 2.7 for cVCA

oocytes; Supplementary Fig. 5B). We hypothesized that this could lead to higher levels of cytoplasmic myosin-II during meiosis I, sterically affecting chromosome capture. We thus visualized active myosin-II at BD + 6 h 30 min, by staining against its phosphorylated light chain pMLC2 (Fig. 5a). Active myosin-II localizes in the cytoplasm and is enriched on the spindle and at the cortex facing the spindle at BD + 6 h 30 min in control oocytes[4,8,16] (Fig. 5a left panels). This staining is specific, as it disappears in oocytes treated with the ML-7 inhibitor, a drug that inhibits myosin-II activation by MLCK phosphorylation[4,8] (Fig. 5a right panels). Active myosin-II is also localized in the cytoplasm and enriched on the spindle at BD + 6 h 30 min in cVCA oocytes (Fig. 5a middle panels). By quantifying active myosin-II fluorescent intensity, we show that the levels of active myosin-II in the cytoplasm and spindle are higher in cVCA oocytes compared with controls (Fig. 5b, c, 1.6 times more active myosin-II in the cytoplasm and 1.4 more in the spindle in cVCA oocytes), reinforcing our hypothesis. Next, we inhibited the

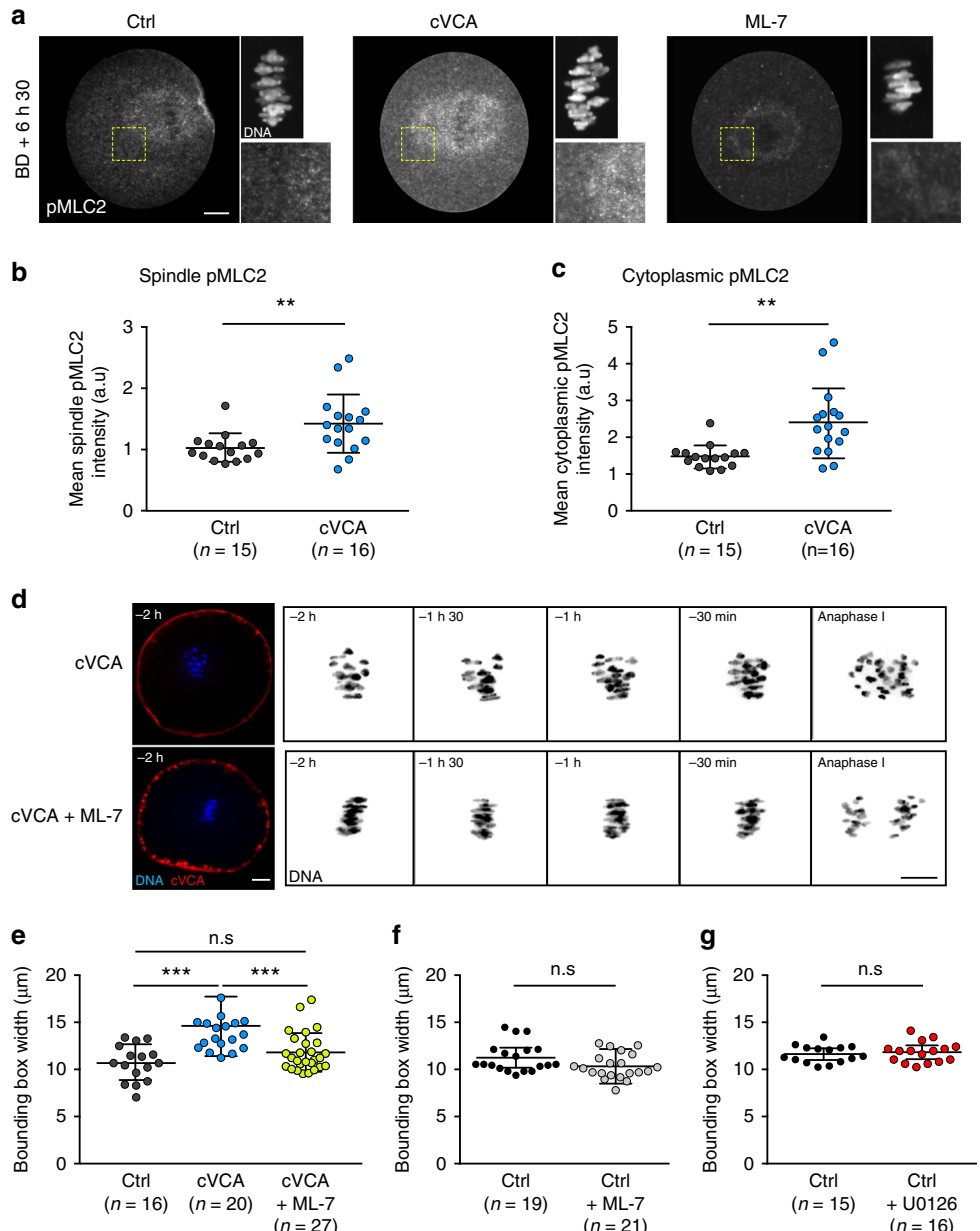

**Fig. 5 Active myosin-II accumulates in extra-soft oocytes and decreasing its activity rescues chromosome misalignment. a** Cropped images of representative control, cVCA, and ML-7-treated oocytes stained for active phosphorylated myosin-II (pMLC2, gray) at BD + 6 h 30 min. Insets are magnification of the chromosomes (gray) and of regions marked by yellow dashed line boxes. Same oocytes as in Fig. 4a. Scale bar: 10 μm. Three independent experiments. **b** Dot plot representing pMLC2 spindle intensity at BD + 6 h 30 min in control and cVCA oocytes. n is the number of oocytes analyzed. Data are from three independent experiments. Statistical significance of differences is assessed with a two-sided Mann–Whitney test: **P-value = 0.0042. **c** Dot plot representing pMLC2 cytoplasm intensity at BD + 6 h 30 min in controls and cVCA oocytes. n is the number of oocytes analyzed. Data are from three independent experiments. Statistical significance of differences is assessed with a two-sided Mann–Whitney test: **P-value = 0.0004. **d** Representative time-lapse movies of cVCA oocytes treated or not with ML-7, expressing Histone(H2B)-GFP (blue or black) and cVCA (red). Acquisitions were taken every 30 min starting 2 h before anaphase I. Scale bar: 10 μm. Six independent experiments. **e** Dot plot representing the metaphase I plate width 30 min before anaphase I in control and cVCA oocytes treated or not with ML-7. n is the number of oocytes analyzed. Data are from six independent experiments. Statistical significance of differences is assessed with a two-sided Mann–Whitney test: n.s. P-value = 0.1708 and ***P-value < 0.0001. **f** Same as **e** for control oocytes treated or not with ML-7. n is the number of oocytes analyzed. Data are from four independent experiments. Statistical significance of differences is assessed with a two-sided t-test: n.s. P-value = 0.0547. **g** Same as **e** for control oocytes treated or not with U0126. n is the number of oocytes analyzed. Data are from three independent experiments. Statistical significance of differences is assessed with a two-sided t-test: n.s. P-value = 0.5756. For **b**, **c**, **e**–**g**, black bars and whiskers represent mean and SD.

activation of myosin-II in extra-soft cVCA oocytes and tested whether it was sufficient to rescue chromosome alignment. cVCA oocytes present chromosome misalignment in metaphase I (Fig. 1b bottom panels, Fig. 2a bottom panels, and Fig. 5d top panels). However, these chromosome defects are no longer observed when they are treated with ML-7: chromosomes are well-aligned on the metaphase I plate until anaphase (Fig. 5d bottom panels). We quantified it by measuring the bounding box

surrounding the metaphase I plate width. The bounding box width is larger in cVCA oocytes (Figs. 5e, 14.61 ± 3.25 μm for cVCA oocytes compared with 10.67 ± 1.90 μm for controls). Remarkably, the bounding box width is significantly rescued in ML-7-treated cVCA oocytes and is comparable to controls (Fig. 5e, 11.81 ± 2.04 μm for cVCA oocytes treated with ML-7). These results argue that an excess of active myosin-II in the cytoplasm impairs chromosome alignment in extra-soft oocytes.

In addition, we tested whether reducing myosin-II activity in control oocytes could improve chromosome alignment, as basal rates of chromosome misalignment (Fig. 1c, g) and aneuploidy (Supplementary Fig. 1) are observed in control populations of mouse oocytes[1]. The bounding box width is reduced in control oocytes treated with ML-7 (Fig. 5f, 11.41 ± 1.58 μm for controls compared with 10.48 ± 1.37 μm for controls treated with ML-7). This trend reinforces the hypothesis that too much active myosin-II in the cytoplasm/spindle hampers chromosome alignment in oocytes.

At last, we quantified chromosome alignment in stiff oocytes. For that, we treated control oocytes with U0126, an inhibitor of the MEK1/2 kinases that mimics a $mos^{-/-}$ phenotype by inhibiting the nucleation of the Arp2/3-dependent subcortex, leading to retention of myosin-II at the cortex and high cortical tension, as we show in a previous study[4]. The bounding box width in U0126-treated oocytes is comparable to controls (Fig. 5g, 11.58 ± 0.92 μm for controls compared with 11.79 ± 1.09 μm for controls treated with U0126), showing that cortical tension per se does not impact chromosome alignment.

## Discussion

In this study, we show that chromosome alignment is impaired by artificial reduction of cortex tension in mouse oocytes (Fig. 1), leading to errors in chromosome segregation and aneuploidy (Supplementary Fig. 1). It is usually thought that changes in cortical tension are transmitted mechanically to the MT spindle. In particular, F-actin is known to impact MT architecture and dynamics, potentially acting on chromosome behavior in oocytes and other systems[9,34–38]. There is also evidence showing that myosin-II is involved in MT functions and thus potentially in chromosome behavior. Myosin-II localizes to the spindle and is implicated in kinetochore-MT flux in metaphase I carne fly spermatocytes[39,40]. It is also localized to chromosome arms and to the spindle in PtK1 cells[41]. In addition, myosin-II is required for proper spindle assembly and positioning in mitotic cells[42] and mouse cardiac myocytes[43]. Another interesting example is the mechanosensitive role of cortical myosin-II on MT growth in endothelial cells, where inhibition of myosin-II activity prevents mitotic centromere-associated kinesin (MCAK)-mediated MT growth[44]. Finally, myosin-II was shown to interact directly with kinesins in astrocytes, which is essential for their migration[45]. Here we show using laser ablation experiments that changes in cortical tension are transmitted mechanically to the MT spindle, but that they do not impact chromosome alignment (Supplementary Fig. 4). We propose that the signal transduction between a strong decrease in cortical tension and chromosome alignment may rather be biochemical. With decreased cortical tension, myosin-II dissociates precociously from the cortex and thus its global concentration increases in the cytoplasm (Fig. 5). This could sterically hinder chromosome capture (Fig. 3), leading to chromosome misalignment and segregation defects. This is reminiscent of starfish oocytes where an actin structure can sterically block kinetochore-MT attachments[32]. Other examples of biochemical signaling between the cortex and the chromosomes are emerging, such as the coordination by Cdk1 of cortical tension maintenance and SAC inactivation at anaphase onset in

mitotic cells[46]. We lack proper tools to investigate precisely how excess of myosin-II presence and/or activity in the cytoplasm could sterically hinder chromosome capture in our model. However, phosphorylation of the regulatory light chain of myosin-II not only increases its ATPase activity but also promotes assembly of myosin-II bipolar thick filaments (for review, see ref. [47]), which could sterically perturb the local capture of kinetochores by MTs. In the future, it would be interesting to address it by computational modeling or using in-vitro models as described in refs. [48,49].

Our results potentially describe an original mode of generation of aneuploidies that could be very common in female gametes. Indeed, 36% of mouse and human oocytes are measured as too soft in a natural population[2] and lower levels of cortical active myosin-II and lower cortical tension are associated with post-ovulatory aging[50]. Thus, some of these naturally soft oocytes could also present chromosome misalignment impeding their future development after fertilization, contributing to the high aneuploidy rate measured in female meiosis[1,51,52]. Measures of cortical mechanical properties could serve as a minimally invasive technique to assess oocyte developmental potential for assisted reproductive technology, as already explored for tumors (for reviews, see refs. [53,54]).

Finally, aberrant cortical tension, and especially cortex softening, are found in a variety of cancer cells[55–58]. Deregulation of myosin-II activity levels has also been described in several human disorders impacting neurons and vessels[59]. Interestingly, overexpression of myosin-II is implicated in cancer progression and metastasis, and myosin-II regulatory pathway genes are increasingly found in disease-associated copy-number variants, particularly in neuronal disorders such as autism and schizophrenia[59], stressing the importance of its regulation. A recent study identified genes involved in mitotic cell rounding, most of which affect cortical myosin-II[60]. The mechanisms described in our study could therefore be relevant for somatic cells in normal and pathological contexts.

## Methods

**Oocyte collection, culture, and microinjection.** Ovaries were collected from 11-week-old OF1 (wt) or $Fmn2^{+/-}$ or $Fmn2^{-/-}$[24] female mice. Fully grown oocytes were extracted by shredding the ovaries[61] in M2 + bovine serum albumin (BSA) medium supplemented with 1 μM milrinone to block and synchronize them in Prophase I[62]. Transferring oocytes into milrinone-free M2 + BSA medium triggers meiosis resumption. All live culture and imaging were carried out under oil at 37 °C.

**Constructs.** We used the following constructs: pRN3-Histone(H2B)-GFP[63], pRN3-hEB3-GFP[29], pspe3-GFP-UtrCH[7], pRN3-SF9-GFP[4,33], pRN3-EzTD-mCherry-VCA[5], pRN3-EzTD-mCherry[5,64], and pTALYM3-TALE- mClover-MajSat[31] (gift from Keith T. Jones, University of Adelaide, Australia).

The pRN3-EzTD-mCherry-FH1FH2 was constructed by cloning a linker GGSGGGSG connected to the FH1FH2 domain of formin 2 (amino acids 734–1578) amplified from a pCS2-FH1FH2-eGFP[13] into pRN3-EzTD-mCherry[5,64].

**In-vitro transcription of cRNAs and microinjection.** Plasmids were linearized using appropriate restriction enzymes. cRNAs were synthesized with the mMessage mMachine kit (Ambion) and subsequently purified using the RNAeasy kit (Qiagen). Their concentration was measured using NanoDrop 2000 from Thermo-Scientific. cRNAs were centrifuged at 4 °C during 45 min prior to microinjection into the cytoplasm of oocytes blocked in Prophase I in M2 + BSA medium supplemented with 1 μM milrinone at 37 °C. cRNAs were microinjected using an Eppendorf Femtojet microinjector[10]. After microinjection, cRNA translation was allowed for 1 or 2 h and oocytes were then transferred into milrinone-free M2 + BSA medium to allow meiosis resumption and meiotic divisions.

**Drug treatments.** ML-7 (Calbiochem, Ref 475880) was diluted at 30 mM in dimethyl sulfoxide (DMSO) and stored at 4 °C. After dilution in M2 medium, it was used on oocytes at 60 μM and added 5 h after NEBD, because at that stage the spindle is bipolar but did not yet migrate to the closest cortex[10]. Control

experiments were conducted in M2 + BSA medium with equivalent concentrations of DMSO.

Nile red stain (Sigma, Ref. N3013) was used to label the total pool of vesicles. It was diluted at 5 mg/ml in DMSO and stored at room temperature. It was used on oocytes at 10 µg/ml.

Cytochalasin D (Life Technologies, Ref. PHZ1063) was diluted at 10 mg/ml in DMSO and stored at −20 °C. It was used on oocytes at 1 µg/ml and added 5 h after NEBD.

Monastrol (Sigma, Ref. M8515) was diluted at 30 mM in DMSO and stored at −20 °C. It was used at 100 µM on oocytes in meiosis I for half an hour starting 1 h 30 min after NEBD and at 200 µM for 2 h on oocytes arrested in meiosis II.

U0126 (Sigma, Ref. 662005) was diluted at 10 mM in DMSO and stored at −20 °C. It was used at 40 µM on oocytes in meiosis I starting 3 h after NEBD[4].

**Immunofluorescence**. After in-vitro culture of oocytes, their zona pellucida was removed by incubation in acid Tyrode's medium (pH 2.3). Oocytes were fixed for 30 min at 37 °C in 4% Formaldehyde (Methanol free) on coverslips treated with gelatin and polylysine. After 20 min of blocking in 0.5% Triton X-100, 3% BSA, antibody staining was performed in phosphate-buffered saline, 0.5% Triton X-100, 3% BSA. As primary antibody, we used rabbit anti-Phospho-MLC2 (Ser19) (Cell Signaling; 1 : 200), at 4 °C overnight. As a secondary antibody, we used Alexa-594-labeled anti-rabbit antibody (Invitrogen; 1 : 400) for 1 h at room temperature. Oocytes were incubated 1 h at room temperature with phalloidin 488 (Invitrogen; 10 U/ml). DNA was stained with Prolong-DAPI (10 µg/ml final 4′,6-diamidino-2-phenylindole (DAPI)).

**Live imaging**. Spinning Disk movies were acquired using (1) a Plan-APO ×40/1.25 NA objective on a Leica DMI6000B microscope enclosed in a thermostatic chamber (Life Imaging Service) equipped with a CoolSnap HQ2/CCD camera coupled to a Sutter filter wheel (Roper Scientific) and a Yokogawa CSU-X1-M1 spinning disk or 2) a Plan-APO ×60/1.4 NA objective on a Ti Nikon microscope enclosed in a thermostatic chamber (Life Imaging Service) equipped with a cMOS camera coupled to a Yokogawa CSU-X1 spinning disk. Metamorph Software (Universal Imaging) was used to collect data.

**Chromosome movement analysis at high temporal resolution**. Oocytes were microinjected in Prophase I with cRNAs encoding Histone(H2B)-GFP with or without cRNAs encoding EzTD-mCherry-VCA to label the chromosomes and decrease or not cortical tension. After microinjection, cRNAs translation was allowed for 1 h. Oocytes were then imaged under a spinning disk at BD + 6 h 30 min. Oocytes were positioned so that their spindle was parallel to the plane of observation and illuminated with an excitation wavelength of 561 nm during 300 ms (first timepoint only) and 491 nm during 300 ms (all timepoints). Acquisitions were taken every 20 s for 20 min, on 3 planes (z-step of 2 µm) focused on the chromosomes. The manual tracking plugin on Fiji (NIH) was used to track chromosome movement and velocity. Only the chromosomes individualized and visible on the three planes for the whole duration of the movie were tracked. For each tracked cluster representing either a chromosome or a group of chromosomes in a given spindle, the squared distances for all possible time steps (dt) were computed and averaged per time step producing an individual MSD value per dt, thus an MSD curve per cluster. All curves corresponding to all clusters of chromosomes were averaged to obtain one MSD curve per spindle. Following this, all curves corresponding to a spindle were averaged per condition to produce one MSD curve per condition.

**Chromosome alignment analysis throughout meiosis I**. Oocytes were microinjected in Prophase I with cRNAs encoding Histone(H2B)-GFP with or without cRNAs encoding EzTD-mCherry-VCA to label the chromosomes and decrease or not cortical tension. After microinjection, cRNAs translation was allowed for 1 h. Oocytes were imaged under the spinning between NEBD and BD + 7 h 30 min. Acquisitions were taken every 30 min on ten z-planes (z-step of 4 µm). Chromosome distribution throughout meiosis I was analyzed in controls and cVCA oocytes. To quantify the repartition of chromosomes, we thresholded the chromosome signal and fitted an ellipse around the binarized images based on image moments in ImageJ. The length of the major axis was given by the eigenvalues of the image moments. We compared the variation of the ellipse lengths during time. For this, we measured the slope of the linear regression of the length over time for a window of 2 h around each timepoint (thus, the slope given at a timepoint $t$ is taken from the values of the ellipsoid length between $t − 1$ h and $t + 1$ h). For each timepoint, we tested for differences of the slope between control and cVCA oocytes spindles with a Wilcoxon's rank test in R.

**Inter major satellites repeats distance**. Oocytes were microinjected in prophase I with cRNAs encoding Histone(H2B)-GFP and MajSat-mClover[65,66] to label the chromosomes and the Major satellite repeats, with or without cRNAs encoding EzTD-mCherry-VCA to decrease or not cortical tension. After microinjection, cRNAs translation was allowed for 1 h. Oocytes were imaged under the spinning disk at BD + 6 h 30 min. Acquisitions were taken on 100 z-planes (z-step of 0.5 µm). For accurate measurements, oocytes were positioned so that their spindle was

parallel to the plane of observation. The distance between major satellites on each bivalent was measured using Metamorph software and used as a proxy to qualitatively assess tension across homologous chromosomes[31].

**Chromosome counting in intact oocytes: monastrol spreads**. Oocytes were microinjected in Prophase I with cRNAs encoding Histone(H2B)-RFP and MajSat-mClover[65,66] to label the chromosomes and the major satellite repeats, with or without cRNAs encoding EzTD-mCherry-VCA to decrease or not cortical tension. Oocytes underwent meiotic maturation in the incubator and arrested in metaphase of meiosis II. At that stage, oocytes were incubated 2 h in 200 µM monastrol, a kinesin-5 inhibitor that causes the bipolar meiosis II spindle to collapse into a monopolar spindle and results in chromosome dispersion[12]. Oocytes were imaged under the spinning disk, with acquisitions spanning the region covering all the chromosomes (z-step of 0.5 µm). The number of chromosomes (MajSat-Clover spots) was counted using the TrackMate plugin in Fiji (http://fiji.sc/TrackMate).

**Oocyte volume measurement using FXm**. The FXm method was initially described in ref. [14] and a detailed protocol is available in ref. [67]. Measurements were made in PDMS chambers that consisted in a simple straight 113 µm high channel. A 2 mm diameter inlet and a 0.5 mm diameter outlet were punched on either side of the channel. The chambers were irreversibly bound to glass-bottomed Petri dishes (Fluorodishes) by plasma treatment. To prevent cell adhesion of the cell and allow reusing the chamber for successive measurements on different cells, chambers were coated with PLL-g-PEG (1%). Before starting the experiment, the chamber was rinsed with M2 + BSA medium containing 0.5 mg/ml of 70 kDa FITC-Dextran (Sigma, Ref FD70S). Oocytes were measured one by one. Each time, the cell was deposited with a mouth-pipette in the inlet. To aspirate the oocyte into the middle of the chamber in a controlled manner, a 250 µl glass syringe was plugged to the outlet via an ~30 cm-long polytetrafluoroethylene (PTFE) tube previously filled with 100 µl of the M2 + BSA dextran solution. The oocyte was then positioned in the center of the chamber and far from the borders of the channel. The chamber was then transferred to a Leica DMIRBE inverted microscope. Bright-field and fluorescence images (excitation wavelength of 491 nm) were acquired using a ×10 NA0.3 objective. For the image analysis, a home-made Matlab software described in ref. [67] was used. Briefly, fluorescent signal was calibrated for every image using the fluorescence intensity ($I_{min}$) under the borders of the chamber ($h_{min} = 0$) and the intensity ($I_{max}$) of the background around the cell ($h_{max} = $ chamber height = 113 µm). The calibration factor $\alpha$ was then calculated as follows: $\alpha = (I_{max} − I_{min})/h_{max}$. The volume of the cell ($V_{cell}$) was obtained by integrating the fluorescence intensity collected under the cell.

$$V_{cell} = \iint_{x,y} \frac{I_{max}(x,y) − I(x,y)}{\alpha} dxdy$$

**Cytoplasmic activity measurements**. The total vesicles stained with Nile red contained in oocytes at BD + 6 h 30 min were imaged every 500 ms with the stream acquisition mode of Metamorph on excitation at 491 nm. Nile Red-labeled vesicle tracking and MSD quantification was performed as described in ref. [15] as follows. After subtracting the background, time-lapse videos were realigned using the rigidbody algorithm of the stackreg plugin in Fiji and then denoised on Metamorph using the Safir denoising program (Roper Scientific). Stacks were then corrected for bleaching using the Histogram Matching algorithm in Fiji and thresholded on Metamorph to generate a binary stack of vesicles. Tracking was then performed on these binary vesicle videos with the TrackMate plugin in Fiji (http://fiji.sc/TrackMate) using DoG detector with a detected object diameter adjusted according to the pattern of vesicles, a thresholding of 1 and sub-pixel localization, and the following settings for gap closing in the simple LAP tracker: linking maximum distance of 1 µm, gap closing maximum distance of 1 µm, and a maximum gap of two frames allowed. Tracks were filtered according to their duration, considering only tracks lasting for more than 25 s. The results of the tracking, including the diameter and the velocities of the vesicles, were provided as Excel and xml files. The velocities values correspond to the mean velocity of a vesicle within a track. For MSD analysis of vesicles trajectories, we used the @msdanalyzer MATLAB class described in: http://bradleymonk.com/matlab/msd/MSDTuto.html. Trajectories were provided in the xml files from Fiji TrackMate analysis.

**Cortical tension measurements**. Cortical tension was measured by micropipette aspiration as described in ref. [4]. Briefly, the zona pellucida of prophase I arrested oocytes was removed by incubating oocytes into M2 + BSA medium supplemented with 0.4% pronase. Oocytes were loaded onto a chamber equilibrated with M2 + BSA medium. A glass micropipette of a diameter five times smaller than the oocyte diameter was connected to a water reservoir of adjustable height to apply a defined aspiration pressure. Zero aspiration pressure was set before each experiment by checking the absence of visible flow inside the pipette. Observations were made through an inverted microscope (Axiovert 200, Zeiss) equipped with a ×40 immersion oil objective (Neofluar 1.3 NA) and connected to a CCD camera (XC-ST70CE, Sony). For every applied pressure, we monitored the length $L$ of the oocyte portion aspirated in the pipette after 30 s and measure at which pressure $L/R = 1$, $R$ being the internal radius of the micropipette. This critical aspiration

pressure $\Delta P_c$ allows calculating the cortical tension $T_c$ of the aspired cell using a viscous drop model[68],

$$T_c = \frac{R \Delta P_c}{2\left(1 - \frac{R}{Rc}\right)}$$

where $Rc$ is the cell radius.

**Microtubule growth speed and density**. Oocytes were microinjected with cRNAs encoding EB3-GFP, with or without cRNAs encoding EzTD-mCherry-VCA. Oocytes were allowed to proceed into prometaphase. At BD + 1 h 30 min, oocytes were incubated in M2 medium supplemented with 100 μM Monastrol for 30 min. MT growth speed was assessed through imaging a single $Z$ plane for 5 s every 250 ms. EB3-GFP comets were tracked with Mosaic plugin for ImageJ[69]. Only the unidirectional tracks with comets visible for ten frames or more were selected and the oocytes included in the analysis had five or more of such tracks.

The monoaster size and brightness were assessed on a maximum brightness projection image from stacks covering the whole volume of the monoaster. The monoaster size was measured by manually fitting an oval around the brightest part of the aster. The fluorescence intensity of EB3-GFP was measured within the central part of the monoaster and normalized by the background fluorescence in the oocyte cytoplasm.

**Laser ablation**. Laser ablation was performed as described in[70]. We used a 355 laser and i-LAS² module (Roper Scientific) coupled to a Leica DMI6000B microscope enclosed in a thermostatic chamber (Life Imaging Service) equipped with a Retiga 3 CCD camera (QImaging) coupled to a Sutter filter wheel (Roper Scientific) and a Yokogawa CSU-X1-M1 spinning disc using a Plan-APO ×40/1.25 NA objective. Metamorph (Universal Imaging) was used to process the data. After calibrating the system, the ablation zone (region of interest, ROI) was determined. It consisted of a line of 15 ±1 μm in length positioned after the spindle pole in the cytoplasm of the oocyte. To perform the laser ablation, the spindle must be oriented with its long axis parallel to the observation plane. The laser ablation parameters used were as follows: 350 nm laser $Z$ thickness 10 μm, dZ 1 μm. Oocytes were then images after ablation with the following parameters: 491 nm laser power during 500 ms, acquisitions every 20 s for 2 min, on one plane. Metaphase plate width was measured by performing bounding boxes containing all the chromosomes with the Fiji (NIH) software. The Microsoft Excel software was used to normalize the data.

**Quantitative image analysis**. Computation of chromosome shapes 30 min before anaphase I: detections were obtained from manually cropped region of interest around the chromosomes using the Phansalkar thresholding method with radius 130 available in Fiji. All pixels of the detection were subsequently used to compute a covariance matrix. Eigenvalues $L_{min}$ and $L_{max}$ were obtained after diagonalization of this matrix and the aspect ratio ($L_{min}/L_{max}$), that is the ratio of the minor axis length to the major axis length, was reported for each cell.

**Quantifications**. - Metaphase plate width was measured on oocytes expressing Histone(H2B)-GFP by performing bounding boxes with the Fiji (NIH) software. Measurements were done 30 minutes before anaphase only on spindles parallel to the imaging plane.
- Myosin-II cortical enrichment was quantified by measuring the ratio of the cortical and cytoplasmic SF9-GFP[33] fluorescence intensities. Oocytes expressed SF9-GFP alone (controls) and together with the cFH1FH2 construct for five hours in Prophase I. The background was removed from all images analyzed using Metamorph software. Cytoplasmic and cortical integrated fluorescence intensities were measured randomly six times in each compartment (cortex and cytoplasm) per oocyte using the same square ROI of 0.12 μm² (region smaller than the cortex width). Measurements were taken on one focal plane (corresponding to the oocyte longest diameter). The cortical fluorescence intensity per square pixels was then divided by the cytoplasmic one.
- For myosin-II cytoplasmic accumulation over time, oocytes expressed SF9-GFP alone (controls) and together with the cFH1FH2 or cVCA constructs for nine hours in Prophase I. The background was removed from all analyzed images using Metamorph software. Integrated fluorescence intensities 2 h and 9 h after cRNA injection were measured randomly six times in the cytoplasm per oocyte using the same square ROI of 80 μm². Measurements were taken on one focal plane (corresponding to the oocyte longest diameter).
- Cortical thickness measurements were performed as described in[4]. Briefly, cortical thickness, consisting of the cortical outer layer (stable throughout meiosis I) and the cortical inner layer (absent in Prophase I and progressively nucleated after NEBD in controls) was measured manually using Fiji software. Oocytes expressed GFP-UtrCH alone (controls) and together with the cFH1FH2 construct for three hours in Prophase I. Four measures per oocyte were randomly taken along the cortex on one focal plane (corresponding to the oocyte longest diameter). The difference between the cytoplasmic and cortical GFP-UtrCH signals was strong enough to discriminate between the two actin networks and to detect the cortical actin network boundary.

- Spindle migration was quantified by measuring distances between the centroid of the oocyte and the centroid of the spindle (oocytes were incubated in SiR-Tubulin at 0.1 μM) using the Fiji (NIH) software. Measuring the distance traveled between the centroid of the spindle relative to the centroid of the oocyte allows discriminating between oocyte movement and spindle movement. Measurements were performed at two timepoints: 4 h before anaphase I when the spindle has not yet started to migrate to the cortex and 30 min before anaphase I when the spindle has almost reached the cortex in the controls. Analyses were done on a Z-projection of 8 stacks of 4 μm, allowing to get the entire spindle. Only spindle parallels to the plane of observation were quantified.
- The spindle length measurements were performed using Metamorph software. Spindle length was measured as the distance between poles in spindles parallel to the imaging plane only.
- The pMLC2 (active phosphorylated myosin-II) and F-actin levels were measured using Metamorph software. Three measures per oocyte were randomly taken within the spindle, and six measures per oocytes were taken within the cytoplasm on a projection of 20 focal planes.

**Statistical analysis**. Experiments were repeated as indicated in the figure legends and a sample of sufficient size was used. The statistical analysis was performed using GraphPad Prism version 8.00 for MacOS, GraphPad Software, La Jolla, CA, USA, www.graphpad.com. For comparisons between two groups, the normality of the variables was checked (D'agostino–Pearson's normality test) and parametric Student's $t$-tests (with Welch's correction when indicated) or non-parametric comparison tests were performed with a confidence interval of 95%. For chromosome alignment experiments, repartitions were analyzed for statistical significance using Fisher's test with a confidence interval of 95%. For volume measurements, a Wilcoxon rank sum test was performed to compare the mean of the two conditions. For the time evolution of SF9-GFP cytoplasmic intensity, a One-way analysis of variance test was performed with a family-wise significance and confidence level of 0.05 (95% confidence interval).

To compare the regression slope values for MSDs, we did a Welch's test of the regression models, as presented in ref. [71] (as the residual variances of the models were not equal).

All error bars are expressed as SD. Values of $P < 0.05$ were considered significant. In all figures, *$P$-value < 0.05, **$P$-value < 0.005, ***$P$-value < 0.0001; n.s., not statistically significant.

**Ethical statement**. All experimental procedures used for the project have been approved by the ministry of agriculture to be conducted in our animal facility (authorization N°75-1170). The use of all the genetically modified organisms described in this project has been granted by the DGRI (Direction Générale de la Recherche et de l'Innovation: Agrément OGM; DUO-1783).

**Reporting summary**. Further information on research design is available in the Nature Research Reporting Summary linked to this article.

## Data availability
All relevant data supporting the finding of this study are available from the corresponding author upon request.

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

## Acknowledgements

We thank Cécile Sykes (Institut Curie, Paris, France) for allowing us to use her micro-pipette aspiration setting and for critical reading of the manuscript. We thank Clément Nizak (ESPCI), Maria Almonacid from the lab and the imaging facility of the CIRB for their input on imaging quantification. We also thank Marie Anfosso and Anthony Gagnon, master students, for their technical help, and Raphaël Voituriez (Sorbonne University, Paris, France) and the members of the Verlhac/Terret team for helpful discussions. We thank Dr. Keith T. Jones, University of Adelaide, Australia, for the MajSat-mClover construct and Greg Fitzharris, University of Montreal, Canada, for sharing his protocol of the Monastrol spreads. This work was supported by grants from the Fondation pour la Recherche Médicale (FRM Label to MHV-DEQ20150331758 then EQU201903007796), from the ANR (ANR-14-CE11 to MHV, ANR-16-CE13 to M.E.T.), and from the Labex Memolife (to M.H.V.). This work has received support from the Fondation Bettencourt Schueller, support under the program « Investissements d'Avenir » launched by the French Government and implemented by the ANR, with the references: ANR-10-LABX-54 MEMO LIFE, ANR-11-IDEX-0001-02 PSL* Research University.

## Author contributions

I.B., M.H.V., and M.E.T. conceived and supervised the project. I.B. performed most experiments. F.C. characterized the cFH1FH2 construct, E.N. did all experiments for the revisions of the manuscript, M.M.S. did the chromosome tracking experiments at high time resolution, they both analyzed their experiments. I.B., M.H.V., and M.E.T. analyzed most experiments. G.L. quantified chromosome behavior throughout meiosis I and computed all MSDs. C. Cadart and R.A. were involved for the volume measurement experiments. C. Campillo was involved for the cortical tension measurement experiments. A.O. and A.G. were involved for the computational 3D imaging approach. A.C. did the original observation that extra-soft oocytes have misaligned chromosomes. I.B. and M.E.T. wrote the manuscript, which was seen and corrected by all authors.

## Competing interests

The authors declare no competing interests.
