## [Peer Review File · Nature Communications]

Reviewers' comments:

Reviewer #1 (Remarks to the Author):

Bennabi and coworkers find in mouse oocytes that compromising the integrity actin cortex interferes with chromosome capture, leading to dramatically increased rates of aneuploidy.

They interfere with the actin cortex by two means: overexpression of a membrane targeted version of the VCA domain activating the Arp2/3 complex (a construct already characterized in previous studies by the same group), and a novel FH1-FH2 domain containing construct inducing fomin-mediated actin nucleation at the cortex. The two different constructs cause very similar effects, which strongly suggest that the effects are general and are due to thickening of the actin cortex.

The authors have shown in previous work that thickening the cortex, somewhat unexpectedly, chases myosin II from the cortex and leads to softening of the cell. Here, the authors show that softening of the oocyte does lead to dramatic shape and even volume changes, but despite these effects spindle assembly and dynamics appears to be unperturbed. Instead, the major defect is seen in chromosome congression and capture. The authors propose a model by which chasing myosin II from the cortex leads to an accumulation of myosin II in the cytoplasm, and apparently this extra cytoplasmic myosin II interferes with chromosome capture. This hypothesis is supported by the experiment in which myosin II is inhibited by ML-7 rescuing the chromosome capture defect. The more detailed mechanism by which myosin II interferes with chromosome capture remains unclear.

The study involves several well designed and rather challenging experiments, which are documented on very clear figures including precise and extensive quantifications. The text is well written, clear and the conclusions are well supported by the data shown. In terms of the findings, these are novel, and likely to have broad relevance in oocytes of diverse species, moreover they are very likely to be relevant as assay for controlling the quality of human oocytes and eggs in assisted reproductive procedures. For these reasons, I highly recommend the publication of the manuscript in Nature Communications. However, I have one major and a few minor concerns, which should be addressed before publication:

Major point:

1. The main conclusion of the manuscript (that is, increased cytoplasmic myosin II activity interferes with chromosome capture) relies on immunostaining by a phospho-myosin antibody and the inhibitor ML-7. Unfortunately, both of these reagents are known in the literature as rather non-specific. For this reason, it would in my opinion be critical to strengthen the main conclusion by additional experiments. Could myosin II be stained by other means? The same group and other labs have used various antibodies and intrabodies to stain myosin II in bulk, for example. Even more importantly, could the (for the conclusions absolutely central) rescue experiment use a reagent other than ML-7 to interfere with myosin II activity? For example, overexpression of myosin phosphatase has been used in other systems (zebrafish), or depletion of myosin II subunits using the Trim21 system recently published by the Schuh lab may be an approach.

Minor points:

1. On Figure 1. It would be important to show the effects of the constructs on the cortex and the cell shape. Therefore, I would move the corresponding panels from the supplement to the main figure. It would be best to show the same kind of quantification for D and H.

2. For the FRAP experiment in Figure 2 it needs to be shown that it is not the turnover of SiR-tubulin on tubulin that is measured. It would be best to repeat the experiment with fluorescently

labeled tubulin.

3. Figure 3 E needs error bars similar to C.

4. Figure 5 C and D: if the two cVCA outliers are removed from the data, is the difference still significant? Likely, more data points are needed.

Reviewer #2 (Remarks to the Author):

In their manuscript, Bennabi et al show that oocytes with reduced cortical tension (extra-soft oocytes) have impaired chromosome alignment. The authors go on to determine that the main cause of chromosomal misalignment is likely to be increased cytoplasmic myosin-II activity. This a very well performed study with high quality data and sensible conclusions. It also has the potential to have broad impact, since oocytes or embryos that are too stiff or too soft are known to fail in development but the mechanism(s) behind these failures remain unclear. However, a major weakness with the current study is whether it truly addresses this broad question, since the vast majority of experiments use a single, artificial, perturbation of the oocyte cortex: expression of cortical VCA (cVCA). I think that the conclusions would be strengthened and have greater novelty if this problem was addressed (see below). Without the suggested additions, I believe the work would be better suited to a more specialist cell biology journal.

Major points:

1. In addition to cVCA, the authors design a new tool to decrease cortical tension: cFH1FH2. Whilst they show that expression of this construct increases misaligned chromosomes, they do very little else with it. Using cFH1FH2 throughout their investigations, to show that their findings aren't limited to just cVCA expression, would help strengthen and broaden the scope of their conclusions.
2. Related to point 1, the authors describe previous studies where soft oocytes are identified as occurring naturally. Whilst I understand that these naturally occurring soft embryos would not be available in the numbers to undertake the mechanistic studies described here (and hence why an artificial means of altering cortical tension is required), surely it would be possible to use naturally soft embryos to validate their findings? In particular, is increased cytoplasmic myosin II activity and chromosome misalignment seen in naturally soft oocytes and can this be rescued by treatment with ML-7?

Minor point:

1. The authors use a novel "aspect ratio" approach to measure chromosome misalignment in Figure 1D. However, they quickly drop this approach in favor of a simpler "bounding box" approach, which is used for the rest of the paper. It isn't clear to me why the aspect ratio approach was used and then dropped. Using the bounding box approach consistently and leaving out the aspect ratio would seem more sensible.

Reviewer #3 (Remarks to the Author):

The presented manuscript studies the effect of changing the cortical tension/stiffness of oocytes upon the behavior of the spindle apparatus. Experiments aimed at understanding spindle behavior and chromosomal aneuploidy are important, as aneuploidy is common in human oocytes, and the study of cytoplasmic structure and actin behavior is topical in terms of spindle positioning, an area in which these authors have been pioneers. A potential link between cortical actin behavior, oocyte cytoplasmic structure, and aneuploidy is therefore of interesting. The authors show that a cVCA construct that reduces cortical tension/stiffness causes delayed chromosome alignment in mouse oocyte maturation. This is accompanied by analyses of the spindle apparatus and chromosomal congression. A model/hypothesis is presented in which myosin that is chased from the cortex to

the central cytoplasm somehow prevents chromosome segregation. Though the idea is interesting, I have the following concerns:

1. The observation that changing oocyte softness impacts chromosome alignment is at least partly predicted by the work of Mogessie and Schuh (Science, 2018). Thus the major novelty here is the data in fig 5 which is interpreted to mean that myosin has a cytoplasmic role in chromosome 'capture'. Much more is needed to prove this point, since there are other alternative explanations. For example, can further fusion proteins be used to specifically modulate cortical vs cytoplasmic and spindle actin? Moreover, the authors need further explanation as to why altered cytoplasmic myosin should alter spindle behavior. Overall more experimental support and explanation is required for the model.
2. Related to the above, the rescue of the cVCA phenotype with ML7 is the most interesting experiment in the paper. Can ML7 prevent errors in a model unrelated to cortical softening, such as aged oocytes, or nocodazole-treated oocytes?
3. Do cVCA and the other treatments cause actual chromosome segregation error? This should be measured with in-situ-chromosome spreads (See Duncan et al., 2009). Without this the misalignment phenotype is hard to interpret.
4. The sirtubulin FRAP experiment should be repeated using another method. The recovery could be the dissociation kinetics of sirtubulin with tubulin, not microtubule dynamics. In addition FRAP (even of tubulin-GFP) is of minimal value in mouse oocytes, where the majority of microtubules are inter-polar. A photo-switchable or photoactivatable tubulin is suggested (See Zhang et al, JCellBiol, 2018).
5. The 'bounding box' alignment analysis is prone to error if the spindle is not precisely aligned in the plane of imaging. In most cases spindles are not labelled, and so the authors cannot be sure of this. At least some of the key experiments need to be repeated with a spindle marker to control for this.
6. A better control for laser ablation is needed than polar bodies to assure there is no damage. Chromosome counts in eggs would be convincing. Also, spindle labelling is necessary to prove the effects seen are not attributable to a change in spindle shape.
7. The authors interpretation of their data makes a lot of assumptions are made about meiosis-I spindle behavior that are not supported by data either here or elsewhere. For example, the assumption that chromosome stretch reflects a certain type of attachment is not clear in the literature. The concept of 'capture' is very vague and not in line with the published literature on mouse oocyte spindle assembly. Lagging chromosomes are a feature of anaphase (not metaphase). By cytoplasmic activity I assume the authors mean kinetic activity

We appreciate the careful review given to our manuscript "Increase in cytoplasmic myosin-II activity impairs chromosome capture in mammalian oocytes". We are pleased that the reviewers were positive about the importance and novelty of our findings and we have sought to address all their concerns as follows. We believe that our revised manuscript is now strengthened. Please note that all changes made to the manuscript are written in red.

Reviewer #1 (Remarks to the Author):

Bennabi and coworkers find in mouse oocytes that compromising the integrity actin cortex interferes with chromosome capture, leading to dramatically increased rates of aneuploidy.

They interfere with the actin cortex by two means: overexpression of a membrane targeted version of the VCA domain activating the Arp2/3 complex (a construct already characterized in previous studies by the same group), and a novel FH1-FH2 domain containing construct inducing fomin-mediated actin nucleation at the cortex. The two different constructs cause very similar effects, which strongly suggest that the effects are general and are due to thickening of the actin cortex.

The authors have shown in previous work that thickening the cortex, somewhat unexpectedly, chases myosin II from the cortex and leads to softening of the cell. Here, the authors show that softening of the oocyte does lead to dramatic shape and even volume changes, but despite these effects spindle assembly and dynamics appears to be unperturbed. Instead, the major defect is seen in chromosome congression and capture. The authors propose a model by which chasing myosin II from the cortex leads to an accumulation of myosin II in the cytoplasm, and apparently this extra cytoplasmic myosin II interferes with chromosome capture. This hypothesis is supported by the experiment in which myosin II is inhibited by ML-7 rescuing the chromosome capture defect. The more detailed mechanism by which myosin II interferes with chromosome capture remains unclear.

The study involves several well designed and rather challenging experiments, which are documented on very clear figures including precise and extensive quantifications. The text is well written, clear and the conclusions are well supported by the data shown. In terms of the findings, these are novel, and likely to have broad relevance in oocytes of diverse species, moreover they are very likely to be relevant as assay for controlling the quality of human oocytes and eggs in assisted reproductive procedures. For these reasons, I highly recommend the publication of the manuscript in Nature Communications. However, I have one major and a few minor concerns, which should be addressed before publication:

We thank Reviewer #1 for such encouraging comments.

Major point:

1. The main conclusion of the manuscript (that is, increased cytoplasmic myosin II activity interferes with chromosome capture) relies on immunostaining by a phospho-myosin antibody and the inhibitor ML-7. Unfortunately, both of these reagents are known in the literature as rather non-specific. For this reason, it would in my opinion be critical to strengthen the main conclusion by additional experiments.

- Could myosin II be stained by other means? The same group and other labs have used various antibodies and intrabodies to stain myosin II in bulk, for example.

We thank the Reviewer for suggesting this experiment that we have done (lines 375-388, new Figure S5) and that strengthens our conclusions.

We quantified myosin-II accumulation over time in the cytoplasm in control and extra soft oocytes (Figure S5). For that, we followed endogenous myosin-II using a specific GFP-coupled intrabody directed against myosin-II (Nizak Traffic 2003; Chaigne Nat Cell Biol 2013; Figure S5A) and measured the fluorescence signal intensity in the cytoplasm 2h and 9h after cRNA injection in control and extra soft oocytes (Figure S5B). Levels of cytoplasmic myosin-II are comparable 2h after cRNA injection in all conditions (control, cVCA, cFH1FH2, Figure S5B).

9h after cRNA injection, extra soft oocytes have 1.9 (for cFH1FH2) and 2.5 (for cVCA oocytes) times more myosin-II in their cytoplasm compared to control oocytes at the same stage (Figure S5B). In addition, extra soft oocytes accumulate more myosin-II in their cytoplasm over time compared to controls (1.5 times accumulation of myosin-II for controls, 2.1 for cFH1FH2 and 2.7 for cVCA oocytes, Figure S5B). Hence, only conditions that allow to extra-soften the oocyte cortex induce a strong increase in cytoplasmic myosin-II levels.

Even more importantly, could the (for the conclusions absolutely central) rescue experiment use a reagent other than ML-7 to interfere with myosin II activity? For example, overexpression of myosin phosphatase has been used in other systems (zebrafish), or depletion of myosin II subunits using the Trim21 system recently published by the Schuh lab may be an approach.

Following the Reviewer's advice, we tried a lot of different means to manipulate myosin-II amount and/or activity.

- First, we used oocytes coming from conditional knockout mice lacking both isoforms encoding myosin-II (Myh9 and Myh10). These animals (Maître Nature 2016; Dumortier Science 2019) were given to us by Jean-Léon Maître (Curie Institute, Paris), whose lab works on early embryonic development in the mouse. As you can see below, oocytes coming from these mice are completely deformed, not amenable to micromanipulations (such as microinjection). In addition, this genotype is extremely difficult to obtain, since one of the strains is subfertile, even when heterozygous level. Thus, we could not work with them.

- Second, we overexpressed myosin phosphatase in oocytes (the constitutively active form Mypt1-T696A used in the zebrafish paper Weiser Dev 2009). The overexpression of CA Mypt1 has no effect and does not trigger any phenotype normally observed when myosin-II is inhibited (absence of spindle migration, absence of cytokinesis, deformation of the plasma membrane).

Images of control oocytes (Control, left panel) and oocytes expressing CA Mypt1 (right panel) arrested in metaphase of meiosis II. Scale bar: 20 μm .

- Third, we tried the reverse, which is to over-activate myosin-II using Calyculin, a Ser/Thr protein phosphatase inhibitor used widely to force myosin-II activation (see list below for few examples of papers, with the doses and model systems):
 Shyer Science 2017, chicken embryo skin (5nM to 25nM).
 Chan Nature 2019, mouse blastocyst embryo (0.5nM).
 Firmino Dev Cell 2016, chicken embryo skin (100nM).

(A) Stills from a time-lapse movie of oocytes treated with 2nM (upper panels) or 3nM (lower panels) Calyculin from BD + 5h. Microtubules (MTs, magenta) are visualized with SIR-Tubulin. Movie starts 6h30 after Nuclear Envelope BreakDown. Acquisitions were taken every 30 minutes. Scale bar: 10 μm .

(B) Images of representative labelling of oocytes stained with an anti- pMLC2 antibody (grey, showing endogenous active phosphorylated myosin-II) at BD + 6h30 for a control (left panel) and an oocyte treated with 2 nM Calyculin (right panel). Scale bar 10 μm .

(C) Dot plot representing the pMLC2 labelling intensity levels at BD + 6h30 in controls (grey) and oocyte treated with 2 nM Calyculin (yellow). (n) is the number of oocytes analyzed. The mean is shown (black bar). Statistical significance of differences is assessed with a Mann-Whitney test: n.s P-value=0.67.

As shown in this figure, treatment of oocytes with 2nM Calyculin has no effect on pMLC2 accumulation (B), showing that at this concentration it does not seem to affect myosin-II activity. However, at 3nM, Calyculin affected mildly the actomyosin cortex (oocytes were slightly deformed, A) but importantly impacted spindle shape inducing a major lengthening (A), as previously observed with global non-specific inhibition of Okaidic acid-sensitive phosphatases in this model system (de Pennart Dev Biol 1993). As such, we could not use this drug on oocytes.

- Fourth, we don't want to use the Trim21 system here because it is based on the specificity of the antibody (and as the referee eluted to earlier the phospho-myosin antibody and all myosin antibodies in general give some non-specific labelling) and it relies on injecting a lot of viscous

material (the antibody), potentially changing the cytoplasmic activity and creating unrelated phenotypes.

All our new attempts to manipulate myosin-II amount and/or activity other than using ML-7, the cVCA, the cFH1FH2 and the U0126 (see below) were not successful. However, to further validate the ML-7, we tested if reducing myosin-II activity in control oocytes could improve chromosome alignment, since basal rates of chromosome misalignment (Figure 1C and G) and aneuploidy (new Figure S1, new Movie S1) are observed in control populations of mouse oocytes (Nagaoka et al., 2012). This experiment is described lines 414-420, new Figure 5F. Strikingly, the bounding box width is reduced in control oocytes treated with ML-7 (Figure 5F, 11.41 ± 1.58 μm for controls compared to 10.48 ± 1.37 μm for controls treated with ML-7). This trend, even if not statistically significant, reinforces the hypothesis that too much active myosin-II in the cytoplasm/spindle hampers chromosome alignment in oocytes even in a control population.

At last, we quantified chromosome alignment in stiff oocytes (lines 421-427, new Figure 5G). For that, we treated control oocytes with U0126, an inhibitor of the MEK1/2 kinases that mimics a *mos*^{-/-} phenotype by inhibiting the nucleation of the Arp2/3-dependent subcortex, leading to retention of myosin-II at the cortex and high cortical tension, as shown previously (Chaigne Nat Cell Biol 2013). The bounding box width in U0126 treated oocytes is comparable to controls (Figure 5G, 11.58 ± 0.92 μm for controls compared to 11.79 ± 1.09 μm for controls treated with U0126), showing that cortical tension per se does not impact chromosome alignment.

Minor points:

1. On Figure 1. It would be important to show the effects of the constructs on the cortex and the cell shape. Therefore, I would move the corresponding panels from the supplement to the main figure. It would be best to show the same kind of quantification for D and H.

We thank the Reviewer for this suggestion. However, due to constraints in figure size and panels, and given the way the paper unfolds, we decided not to do so.

With respect to the quantifications in Figure 1D and H, we apologize for not having better expressed what they were used for, as Reviewer #2 also noted. It was important for us to have a blind approach to discriminate between controls and extra-soft oocytes, and to determine which criteria showed the greatest statistical difference between these two populations. This was achieved using an unbiased computational imaging approach to automatically threshold the stacks of images and extract the features differing the most between controls and extra-soft oocytes (as in Almonacid Dev Cell 2019). Using this approach, the two principal features that showed the greatest statistical difference between these two populations of oocytes were chromosome alignment and cell shape. For chromosome alignment, it was the bounding box measure. This tedious approach blindly validated the defects we could observe visually, and allowed us to find a criterion to measure them in an unbiased manner (Figure 1D). This criterion was then applied manually in the rest of the paper by drawing the bounding boxes around the chromosomes (Figure 1H). At last, Control and cVCA oocytes were measured the same way as Figure 1H in Figure 5E. We modified the text accordingly (lines 123-128 and 137-142).

2. For the FRAP experiment in Figure 2 it needs to be shown that it is not the turnover of SiR-tubulin on tubulin that is measured. It would be best to repeat the experiment with fluorescently labeled tubulin.

We agree with the Reviewer that the recovery could be the dissociation kinetics of Sir-tubulin with tubulin and not microtubule dynamics, point also raised by Reviewer #3. Reviewer #3 also said that FRAP (even of tubulin-GFP) is of minimal value in mouse oocytes, where the majority of microtubules are inter-polar. As such we did not repeat the experiments with fluorescently labeled tubulin because we knew that Reviewer #3 would dismiss the results. Reviewer #3 suggested a photo-switchable or photoactivatable tubulin. However, this technique was only used in mouse oocytes arrested in meiosis II (Fitzharris Dev 2009; Fitzharris Curr Biol 2012; Mogessie Science 2018). It appears that it is not fully operational in early meiosis I oocytes because the spindle forms deeper in the cell compared to meiosis II where it is subcortical.

In an attempt to address the concerns of both Reviewers, we measured relative MT densities and MT growth early on during spindle morphogenesis since defects at this stage are known to induce chromosome mis-alignment later on (for review see Bennabi J Cell Biol 2016; Bennabi EMBO Rep 2018; Letort Mol Cell Biol 2019). This experiment is described in lines 290-298, new Figure 2B-C. Control and cVCA oocytes expressing EB3-GFP, a MT plus-end tracker, were imaged at BD + 2h after monastrol treatment to inhibit spindle bipolarization (as in Breuer J Cell Biol 2010; new Movie S9). MT densities and monoaster sizes are comparable in control (grey dots) and extra-soft (blue dots) oocytes (Figure 2B dot plots and yellow dashed circles). Tracking of individual MT plus-ends (Figure 2C red tracks) show that MT growth rates are also comparable in control (grey dots) and extra-soft oocytes (blue dots, Figure 2C). Hence, consistent with our observations using FRAP on sir-Tubulin, we do not seem to observe major impact of cortex softening on MTs dynamics.

3. Figure 3 E needs error bars similar to C.

We cannot represent the error bars in Figure 3E since they are percentages. However, the statistics (Chi-square test) are written in the legend.

4. Figure 5 C and D: if the two cVCA outliers are removed from the data, is the difference still significant? Likely, more data points are needed.

Please find below the dot plots corresponding to Figure 5B and C without the two cVCA outliers (A) and without the two cVCA outliers and the Ctrl outlier (B). The difference is still significant in all cases.

Dot plots representing the pMLC2 spindle intensity levels at BD + 6h30 in control (black) and cVCA oocytes (blue). (n) is the number of oocytes analyzed. Data are from 3 independent experiments. The mean is shown (black bar). Statistical significance of differences is assessed with Mann-Whitney tests. Corresponds to Figure 5B and C without the 2 cVCA outliers.

B

Dot plots representing the pMLC2 spindle intensity levels at BD + 6h30 in control (black) and cVCA oocytes (blue). (n) is the number of oocytes analyzed. Data are from 3 independent experiments. The mean is shown (black bar). Statistical significance of differences is assessed with Mann-Whitney tests. Corresponds to Figure 5B and C without the 2 cVCA outliers and the Ctrl outlier.

As such, we left the original dot plot in the paper.

Reviewer #2 (Remarks to the Author):

In their manuscript, Bennabi et al show that oocytes with reduced cortical tension (extra-soft oocytes) have impaired chromosome alignment. The authors go on to determine that the main cause of chromosomal misalignment is likely to be increased cytoplasmic myosin-II activity. This a very well performed study with high quality data and sensible conclusions. It also has the potential to have broad impact, since oocytes or embryos that are too stiff or too soft are known to fail in development but the mechanism(s) behind these failures remain unclear. However, a major weakness with the current study is whether it truly addresses this broad question, since the vast majority of experiments use a single, artificial, perturbation of the oocyte cortex: expression of cortical VCA (cVCA). I think that the conclusions would be strengthened and have greater novelty if this problem was addressed (see below). Without the suggested additions, I believe the work would be better suited to a more specialist cell biology journal.

We thank Reviewer # 2 for his/her thoughtful comments.

Major points:

1. In addition to cVCA, the authors design a new tool to decrease cortical tension: cFH1FH2. Whilst they show that expression of this construct increases misaligned chromosomes, they do very little else with it. Using cFH1FH2 throughout their investigations, to show that their findings aren't limited to just cVCA expression, would help strengthen and broaden the scope of their conclusions.

First, we would like to highlight the fact that we had to characterize from scratch the cFH1FH2 (a whole supplemental figure is dedicated to it), and show that it phenocopies the cVCA phenotype (misaligned chromosomes).

To go with the Reviewer advice, we quantified myosin-II accumulation over time in the cytoplasm in control and extra soft oocytes (Figure S5). For that, we followed endogenous myosin-II using a specific GFP-coupled intrabody directed against myosin-II (Nizak Traffic 2003; Chaigne Nat Cell Biol 2013; Figure S5A) and measured the fluorescence signal intensity in the cytoplasm 2h and 9h after cRNA injection in control and extra soft oocytes (Figure S5B). Levels of cytoplasmic myosin-II are comparable 2h after cRNA injection in all conditions (control, cVCA, cFH1FH2, Figure S5B). 9h after cRNA injection, extra soft oocytes have 1.9 (for cFH1FH2) and 2.5 (for cVCA oocytes) times more myosin-II in their cytoplasm compared

to control oocytes at the same stage (Figure S5B). In addition, extra soft oocytes accumulate more myosin-II in their cytoplasm over time compared to controls (1.5 times accumulation of myosin-II for controls, 2.1 for cFH1FH2 and 2.7 for cVCA oocytes, Figure S5B). Hence, only conditions that allow to extra-soften the oocyte cortex induce a strong increase in cytoplasmic myosin-II levels.

2. Related to point 1, the authors describe previous studies where soft oocytes are identified as occurring naturally. Whilst I understand that these naturally occurring soft embryos would not be available in the numbers to undertake the mechanistic studies described here (and hence why an artificial means of altering cortical tension is required), surely it would be possible to use naturally soft embryos to validate their findings? In particular, is increased cytoplasmic myosin II activity and chromosome misalignment seen in naturally soft oocytes and can this be rescued by treatment with ML-7?

We fully agree with the Reviewer that the ultimate goal would be to establish a direct link in a single oocyte coming from a natural oocyte population between cortical tension defects (extra-soft) and chromosome misalignment. What the Reviewer is asking for is not easy: it is to measure and identify prophase oocytes that are too soft in a normal population, then follow them until anaphase and analyze the alignment and separation of their chromosomes (with or without ML-7). This is a project on its own that we are just starting, designing new devices and applying new technologies (such as Atomic Force Microscopy) to address this question in a high throughput manner and to set threshold values of cortical tension (which we do not know at the moment) that correlate with a certain type of phenotype. Thus, we believe it is beyond the scope of this paper.

However, we tested if reducing myosin-II activity in control oocytes could improve chromosome alignment, since basal rates of chromosome misalignment (Figure 1C and G) and aneuploidy (new Figure S1, new Movie S1) are observed in control populations of mouse oocytes (Nagaoka et al., 2012). This experiment is described lines 414-420, new Figure 5F. Strikingly, the bounding box width is reduced in control oocytes treated with ML-7 (Figure 5F, 11.41 ± 1.58 μm for controls compared to 10.48 ± 1.37 μm for controls treated with ML-7). This trend, even if not statistically significant, reinforces the hypothesis that too much active myosin-II in the cytoplasm/spindle hampers chromosome alignment in oocytes even in a control population.

At last, we quantified chromosome alignment in stiff oocytes (lines 421-427, new Figure 5G). For that, we treated control oocytes with U0126, an inhibitor of the MEK1/2 kinases that mimics a *mos*^{-/-} phenotype by inhibiting the nucleation of the Arp2/3-dependent subcortex, leading to retention of myosin-II at the cortex and high cortical tension, as shown previously (Chaigne Nat Cell Biol 2013). The bounding box width in U0126 treated oocytes is comparable to controls (Figure 5G, 11.58 ± 0.92 μm for controls compared to 11.79 ± 1.09 μm for controls treated with U0126), showing that cortical tension per se does not impact chromosome alignment.

Minor point:

1. The authors use a novel "aspect ratio" approach to measure chromosome misalignment in Figure 1D. However, they quickly drop this approach in favor of a simpler "bounding box" approach, which is used for the rest of the paper. It isn't clear to me why the aspect ratio approach was used and then dropped. Using the bounding box approach consistently and leaving out the aspect ratio would seem more sensible.

With respect to the quantifications in Figure 1D and H, we apologize for not having better expressed what they were used for, as Reviewer #1 also noted. It was important for us to have a blind approach to discriminate between controls and extra-soft oocytes, and to determine

which criteria showed the greatest statistical difference between these two populations. This was achieved using an unbiased computational imaging approach to automatically threshold the stacks of images and extract the features differing the most between controls and extra-soft oocytes (as in Almonacid Dev Cell 2019). Using this approach, the two principal features that showed the greatest statistical difference between these two populations of oocytes were chromosome alignment and cell shape. For chromosome alignment, it was the bounding box measure. This tedious approach blindly validated the defects we could observe visually, and allowed us to find a criterion to measure them in an unbiased manner (Figure 1D). This criterion was then applied manually in the rest of the paper by drawing the bounding boxes around the chromosomes (Figure 1H). At last, Control and cVCA oocytes were measured the same way as Figure 1H in Figure 5E. We modified the text accordingly (lines 123-128 and 137-142).

Reviewer #3 (Remarks to the Author):

The presented manuscript studies the effect of changing the cortical tension/stiffness of oocytes upon the behavior of the spindle apparatus. Experiments aimed at understanding spindle behavior and chromosomal aneuploidy are important, as aneuploidy is common in human oocytes, and the study of cytoplasmic structure and actin behavior is topical in terms of spindle positioning, an area in which these authors have been pioneers. A potential link between cortical actin behavior, oocyte cytoplasmic structure, and aneuploidy is therefore of interesting. The authors show that a cVCA construct that reduces cortical tension/stiffness causes delayed chromosome alignment in mouse oocyte maturation. This is accompanied by analyses of the spindle apparatus and chromosomal congression. A model/hypothesis is presented in which myosin that is chased from the cortex to the central cytoplasm somehow prevents chromosome segregation. Though the idea is interesting, I have the following concerns:

1. The observation that changing oocyte softness impacts chromosome alignment is at least partly predicted by the work of Mogessie and Schuh (Science, 2018).

We respectfully disagree with this statement. The work of Mogessie and Schuh (Science 2018) shows that actin in the cytoplasm drives K-fibers formation (using oocytes depleted of cytoplasmic actin). Despite the presence of K-fibers in the spindle before anaphase, the phenotype observed in this paper is lagging chromosomes at anaphase I, never chromosome misalignment in meiosis I.

Here, we explore the effect of changes in oocyte stiffness on chromosome alignment (not explored at all in the Mogessie and Schuh (Science 2018) work). We show that cytoplasmic actin is not affected in extra soft oocytes (Chaigne Nat Commun 2015 and Figure 4A- C). The phenotype we observe is occurring earlier and is different from the one they observe through alteration of K-fibers formation: we see chromosome misalignment in meiosis I, leading to aneuploidy (see below response to point 3). This reinforces the fact that these phenotypes (lagging chromosomes in the Mogessie paper and chromosome misalignment in our paper) do not have a common origin, and that we are not modifying K-fiber formation here (otherwise why would we see a precocious phenotype in metaphase compared to anaphase in their paper)?

Thus the major novelty here is the data in fig 5 which is interpreted to mean that myosin has a cytoplasmic role in chromosome 'capture'. Much more is needed to prove this point, since there are other alternative explanations. For example, can further fusion proteins be used to specifically modulate cortical vs cytoplasmic and spindle actin?

To precise our point, Figure 5 is not intended to mean that myosin has a physiological cytoplasmic role in chromosome "capture" but that too much myosin perturbs chromosome "capture".

We believe that between our papers and others, plus the experiment we added (U0126, see below and response to point 2), the effect of specific modulation of cortical vs cytoplasmic actin on chromosome alignment have been addressed. Here is a table summarizing the literature and our experiments:

	Cytoplasmic actin meshwork	Spindle actin	Subcortical actin network	Cortical tension	Spindle morphogenesis	Chromosome alignment in MI
Formin2 knockout oocytes, oocytes treated with cytochalasin D (Chaigne NCB 2013; Mogessie Science 2018; our paper)	Absent	Absent	Normal	Normal	Normal	Normal
mos knockout oocytes, oocytes treated with U0126 (Verlhac Curr Biol 2000; Chaigne NCB 2013; our paper)	Normal	Normal	Absent	Stiff	Normal	Normal
cVCA and cFH1FH2 oocytes (Chaigne Nat Comm 2015; our paper)	Normal	Normal	Precocious	Soft	Normal	Misalignment
MAP4-UtrCH (Mogessie Science 2018)	Normal	Too much	unknown	unknown	Aberrant	Misalignment
SIR-Actin, Fmn2 OE (Azoury Dev 2011; Mogessie Science 2018)	Too much	Too much	unknown	unknown	Aberrant	Misalignment

What we and others never managed to do (not for the lack of having tried by different means: laser ablation, targeting of actin severing proteins to the spindle...) is depleting actin specifically in the spindle. Thus, I don't see what more we could try, and what it would add to the paper.

Moreover, the authors need further explanation as to why altered cytoplasmic myosin should alter spindle behavior. Overall more experimental support and explanation is required for the model.

Our hypothesis is that too much myosin-II interferes with chromosome capture, not via a specific binding to the kinetochore (even if binding of myosin-II to chromosomes was described in the literature: Robinson Protoplasma 2005), but rather by steric hindrance. As we said in the discussion of the paper, we lack proper tools to investigate precisely how an excess of myosin-II presence and/or activity in the cytoplasm could sterically hinder chromosome capture in our model (mouse oocyte). However, phosphorylation of the regulatory light chain of myosin-II not only increases its ATPase activity but also promotes assembly of myosin-II bipolar thick filaments (for review see Shutova BBRC 2009), that could sterically perturb the local capture of kinetochores by microtubules. In the future, it would be interesting to address it by computational modeling or using *in vitro* models as described in Reymann Science 2012 and Ennomani Curr Biol 2016. At present, a lot of quantitative data are missing, enabling proper feeding for these *in vitro* and *in silico* studies, that we believe are beyond the scope of this paper. Despite this, we are glad to see that all the Reviewers are clear that the achievements of the paper constitute a major advance.

In addition, we did a lot of experiments to strengthen the paper:

1/ We quantified myosin-II accumulation over time in the cytoplasm in control and extra soft oocytes (lines 375-388, new Figure S5). For that, we followed endogenous myosin-II using a specific GFP-coupled intrabody directed against myosin-II (Nizak Traffic 2003; Chaigne Nat Cell Biol 2013; Figure S5A) and measured the fluorescence signal intensity in the cytoplasm

2h and 9h after cRNA injection in control and extra soft oocytes (Figure S5B). Levels of cytoplasmic myosin-II are comparable 2h after cRNA injection in all conditions (control, cVCA, cFH1FH2, Figure S5B). 9h after cRNA injection, extra soft oocytes have 1.9 (for cFH1FH2) and 2.5 (for cVCA oocytes) times more myosin-II in their cytoplasm compared to control oocytes at the same stage (Figure S5B). In addition, extra soft oocytes accumulate more myosin-II in their cytoplasm over time compared to controls (1.5 times accumulation of myosin-II for controls, 2.1 for cFH1FH2 and 2.7 for cVCA oocytes, Figure S5B). Hence, only conditions that allow to extra-soften the oocyte cortex induce a strong increase in cytoplasmic myosin-II levels.

2/ We tried a lot of different means to manipulate myosin-II amount and/or activity.

- First, we used oocytes coming from conditional knockout mice lacking both isoforms encoding myosin-II (Myh9 and Myh10). These animals (Maître Nature 2016; Dumortier Science 2019) were given to us by Jean-Léon Maître (Curie Institute, Paris), whose lab works on early embryonic development in the mouse. As you can see below, oocytes coming from these mice are completely deformed, not amenable to micromanipulations (such as microinjection). In addition, this genotype is extremely difficult to obtain, since one of the strains is subfertile, even when heterozygous level. Thus, we could not work with them.

- Second, we overexpressed myosin phosphatase in oocytes (the constitutively active form Mypt1-T696A used in the zebrafish paper Weiser Dev 2009). The overexpression of CA Mypt1 has no effect and does not trigger any phenotype normally observed when myosin-II is inhibited (absence of spindle migration, absence of cytokinesis, deformation of the plasma membrane).

Images of control oocytes (Control, left panel) and oocytes expressing CA Mypt1 (right panel) arrested in metaphase of meiosis II. Scale bar: 20 μ m.

- Third, we tried the reverse, which is to over-activate myosin-II using Calyculin, a Ser/Thr protein phosphatase inhibitor used widely to force myosin-II activation (see list below for few examples of papers, with the doses and model systems):
 Shyer Science 2017, chicken embryo skin (5nM to 25nM).
 Chan Nature 2019, mouse blastocyst embryo (0.5nM).
 Firmino Dev Cell 2016, chicken embryo skin (100nM).

(A) Stills from a time-lapse movie of oocytes treated with 2nM (upper panels) or 3nM (lower panels) Calyculin from BD + 5h. Microtubules (MTs, magenta) are visualized with SIR-Tubulin. Movie starts 6h30 after Nuclear Envelope BreakDown. Acquisitions were taken every 30 minutes. Scale bar: 10 μ m.
 (B) Images of representative labelling of oocytes stained with an anti- pMLC2 antibody (grey, showing endogenous active phosphorylated myosin-II) at BD + 6h30 for a control (left panel) and an oocyte treated with 2 nM Calyculin (right panel). Scale bar 10 μ m.
 (C) Dot plot representing the pMLC2 labelling intensity levels at BD + 6h30 in controls (grey) and oocyte treated with 2 nM Calyculin (yellow). (n) is the number of oocytes analyzed. The mean is shown (black bar). Statistical significance of differences is assessed with a Mann-Whitney test: n.s P-value=0.67.

As shown in this figure, treatment of oocytes with 2nM Calyculin has no effect on pMLC2 accumulation (B), showing that at this concentration it does not seem to affect myosin-II activity. However, at 3nM, Calyculin affected mildly the actomyosin cortex (oocytes were slightly deformed, A) but importantly impacted spindle shape inducing a major lengthening (A), as previously observed with global non-specific inhibition of Okaidic acid-sensitive phosphatases in this model system (de Pennart Dev Biol 1993). As such, we could not use this drug on oocytes.

All our new attempts to manipulate myosin-II amount and/or activity other than using ML-7, the cVCA, the cFH1FH2 and the U0126 (see below) were not successful. I don't see what more we could try. However, we managed to further validate the ML-7, see our response to point 2 below.

2. Related to the above, the rescue of the cVCA phenotype with ML7 is the most interesting experiment in the

paper. Can ML7 prevent errors in a model unrelated to cortical softening, such as aged oocytes, or nocodazole-treated oocytes?

To answer to the Reviewer question and further validate the ML-7, we tested if reducing myosin-II activity in control oocytes could improve chromosome alignment, since basal rates of chromosome misalignment (Figure 1C and G) and aneuploidy (new Figure S1, new Movie S1) are observed in control populations of mouse oocytes (Nagaoka et al., 2012). This experiment is described lines 414-420, new Figure 5F. Strikingly, the bounding box width is reduced in control oocytes treated with ML-7 (Figure 5F, 11.41 ± 1.58 μm for controls compared to 10.48 ± 1.37 μm for controls treated with ML-7). This trend, even if not statistically significant, reinforces the hypothesis that too much active myosin-II in the cytoplasm/spindle hampers chromosome alignment in oocytes even in a control population. At last, we quantified chromosome alignment in stiff oocytes (lines 421-427, new Figure 5G). For that, we treated control oocytes with U0126, an inhibitor of the MEK1/2 kinases that mimics a *mos*^{-/-} phenotype by inhibiting the nucleation of the Arp2/3-dependent subcortex, leading to retention of myosin-II at the cortex and high cortical tension, as shown previously (Chaigne Nat Cell Biol 2013). The bounding box width in U0126 treated oocytes is comparable to controls (Figure 5G, 11.58 ± 0.92 μm for controls compared to 11.79 ± 1.09 μm for controls treated with U0126), showing that cortical tension per se does not impact chromosome alignment.

3. Do cVCA and the other treatments cause actual chromosome segregation error? This should be measured with in-situ-chromosome spreads (See Duncan et al., 2009). Without this the misalignment phenotype is hard to interpret.

We thank the Reviewer for his/her suggestion that brings the paper to another level. We show now (lines 119-123, new Figure S1, new Movie S1) that chromosome misalignment in cVCA oocytes leads to aberrant chromosome segregation at anaphase I (Figure 1B, red asterisk) and aneuploidy (Figure S1, 80% of oocytes are euploid, containing 20 chromosomes in controls versus 37.5% in cVCA oocytes, Movie S1), measured in intact oocytes using Monastrol spreads (see Material and Methods; Duncan Curr Biol 2009).

4. The sirtubulin FRAP experiment should be repeated using another method. The recovery could be the dissociation kinetics of sirtubulin with tubulin, not microtubule dynamics. In addition FRAP (even of tubulin-GFP) is of minimal value in mouse oocytes, where the majority of microtubules are inter-polar. A photo-switchable or photoactivatable tubulin is suggested (See Zhang et al, JCellBiol, 2018).

We agree with the Reviewer that the recovery could be the dissociation kinetics of Sir-tubulin with tubulin and not microtubule dynamics, point also raised by Reviewer #1. As you said that FRAP (even of tubulin-GFP) is of minimal value in mouse oocytes, where the majority of microtubules are inter-polar. As such we did not repeat the experiments with fluorescently labeled tubulin because we knew that Reviewer #3 would dismiss the results. Reviewer #3 suggested a photo-switchable or photoactivatable tubulin. However, this technique was only used in mouse oocytes arrested in meiosis II (Fitzharris Dev 2009; Fitzharris Curr Biol 2012; Mogessie Science 2018). It appears that it is not fully operational in early meiosis I oocytes because the spindle forms deeper in the cell compared to meiosis II where it is subcortical.

In an attempt to address the concerns of both Reviewers, we measured relative MT densities and MT growth early on during spindle morphogenesis since defects at this stage are known to induce chromosome mis-alignment later on (for review see Bennabi J Cell Biol 2016; Bennabi EMBO Rep 2018; Letort Mol Cell Biol 2019). This experiment is described in lines 290-298, new Figure 2B-C. Control and cVCA oocytes expressing EB3-GFP, a MT plus-end tracker,

were imaged at BD + 2h after monastrol treatment to inhibit spindle bipolarization (as in Breuer J Cell Biol 2010; new Movie S9). MT densities and monoaster sizes are comparable in control (grey dots) and extra-soft (blue dots) oocytes (Figure 2B dot plots and yellow dashed circles). Tracking of individual MT plus-ends (Figure 2C red tracks) show that MT growth rates are also comparable in control (grey dots) and extra-soft oocytes (blue dots, Figure 2C). Hence, consistent with our observations using FRAP on α -Tubulin, we do not seem to observe major impact of cortex softening on MTs dynamics.

5. The 'bounding box' alignment analysis is prone to error if the spindle is not precisely aligned in the plane of imaging. In most cases spindles are not labelled, and so the authors cannot be sure of this. At least some of the key experiments need to be repeated with a spindle marker to control for this.

We thank the referee for voicing his/her concern, but having, with others, being a pioneer in the field of spindle morphogenesis in mouse oocytes, we are fully aware of this problem and consider only the spindles parallel to the plane of observation (quantifications were done with spindle markers not necessarily visible on the images see response to point 6, or by setting a maximum size in term of number of z-planes for the metaphase plate). The fact that we consider and know how to recognize only the spindles parallel to the plane of observation is further highlighted by the measure of the spindle length in control and cVCA oocytes (Figure 2D, comparable size) and control and cFH1FH2 oocytes (lines 159-163, new Figure S2F, comparable size).

6. A better control for laser ablation is needed than polar bodies to assure there is no damage. Chromosome counts in eggs would be convincing. Also, spindle labelling is necessary to prove the effects seen are not attributable to a change in spindle shape.

We don't see the necessity of doing such a control (spreads in meiosis II after laser ablation in meiosis I). We are analyzing the immediate response of laser ablation on a very short time scale (1 minute after ablation) and the polar body control was meant to show that we are not looking at dying oocytes after ablation. We are not looking at the effect of laser ablation on chromosome alignment and segregation 2h after ablation.

We actually always do laser ablation in the presence of spindle labelling (very low doses of EB3-GFP), as explained in the manuscript (line 230 in the previous version). The signal is barely visible in new Figure S4 (old figure S3) because if we want to visualize the spindle, the chromosomes appear saturated (see below, A). We always label the spindle to make sure that the ablation is not performed within the spindle but away from spindle poles. The spindle pole closest to the site of laser ablation is bleaching after ablation (see below, A), so we cannot measure full spindle length after laser ablation. However, we can follow the evolution in length of the hemi-spindle away from the ablation point after laser ablation (see below, B) and we show here that they are comparable in control and cVCA oocytes (see below, B).

(A) Stills from a time-lapse movie of a control oocyte (upper panel) and cVCA oocyte (lower panel) expressing Histone(H2B)-GFP to label chromosomes and EB3-GFP to label the spindle (both in grey) at BD + 6h30. First timepoint before laser ablation, then acquisitions taken every 20 seconds during 1 minute. Scale bar: 10 μ m. Note that the contrast is enhanced so that the spindle is visible at all timepoints, especially the hemi-spindle on the right side of the metaphase plate that bleaches after ablation. Same images as in new Figure S4.

(B) Graph representing the relative decrease in length of the hemi-spindle away from the ablation zone, observed every 20 seconds (s) during 1 minute in control (black) and cVCA oocytes (blue). (n) is the number of oocytes analyzed. Data are compiled from 5 independent experiments. The mean and standard deviation (SD) are shown for each timepoint. The hemi-spindle length was normalized so that 1 corresponds to the length before ablation at t=0 seconds.

7. The authors interpretation of their data makes a lot of assumptions are made about meiosis-I spindle behavior that are not supported by data either here or elsewhere. For example, the assumption that chromosome stretch reflects a certain type of attachment is not clear in the literature.

We apologize if we were not clear. We use the distance between Major satellite repeats as an indirect measure of tension across bivalents, not their attachment, as a lot of other labs do in oocytes: Kitajima lab (Sakakibara Nat Comm 2015; Yoshida Dev Cell 2015), Hoog lab (Kouznetsova Nat Comm 2014; Kouznetsova EMBO Rep 2019), Ellenberg lab (Kitajima 2011; Solc Plos One 2015), Jones lab (Lane Dev 2012 ; Wu Nat Comm 2018), Wassmann lab (Vallot Curr Biol 2018), Herbert lab (Lister Curr Biol 2010), Schuh lab (Zielinska ELife 2015), Lampson lab (Chiang Curr Biol 2010), our lab (Kolano PNAS 2012). For more clarity on this issue, we have removed the 2 occurrences in our manuscript where the terms attachment (for the chromosomes, line 325 of the previous version of the manuscript) or bioriented (line 975 of the previous version of the manuscript) were used.

The concept of 'capture' is very vague and not in line with the published literature on mouse oocyte spindle assembly.

Could the Reviewer be more specific, develop and give precise examples? We don't really know how to answer this comment in the absence of any specific recommendation. As discussed in our paper, this concept is in line with the Kitajima Cell 2011 work which is widely used as a reference in the field.

Lagging chromosomes are a feature of anaphase (not metaphase).

Thank you for noticing, we have deleted the only occurrence in line 112 of the previous version of the manuscript where it was written.

By cytoplasmic activity I assume the authors mean kinetic activity

We are not sure to fully understand the question here. Cytoskeletal dynamics impacts the

surrounding cytoplasm by dragging it along and generating cytoplasmic motion. Here cytoplasmic activity represents the movement of the cytoplasm within the cell.

REVIEWERS' COMMENTS:

Reviewer #1 (Remarks to the Author):

The authors made major attempts to address the criticism raised by me and the other reviewers. These clarified important points and significantly strengthened the conclusions. Overall, I would be happy to support the publication of this revised version in Nature Communications. However, I would have two additional remarks:

1. While the authors convincingly show in their rebuttal letter that they have made any reasonable attempt to more directly address the role of myosin II, unfortunately all these attempts failed. Overall, the findings are important and are now supported by various lines of indirect evidence, for which reason I am in support of publication. However, the discussion should still be formulated carefully to point out the lack of direct evidence in support of this main conclusion.
2. Especially for the revised sections, the phrasing is rather imprecise at certain places. For example, by saying "This tedious computational approach blindly quantified..." they probably mean that the approach is rigorous and is unbiased. Similarly, at a later point, they appear to draw a conclusion from statistically non-significant differences. They should provide an explanation for this. Maybe the statistical test was not appropriate?

Reviewer #2 (Remarks to the Author):

I am impressed by how well the authors have addressed my concerns. By adding in additional studies with cFH1FH2 they convincingly show similar results with cVCA and cFH1FH2, most crucially now showing increased cytoplasmic myosin II with cFH1FH2 just as with cVCA (FigS5). I also appreciate that my suggestion to investigate naturally occurring soft oocytes was a big ask but I am happy that authors went some way to address this by reducing myosin II activity in control embryos and finding a slight improvement in chromosome alignment (Fig5). I am therefore happy to recommend publication of the revised manuscript in Nature Communications.

Reviewer #3 (Remarks to the Author):

The authors have made considerable effort to improve the manuscript in line with reviewers comments. The following two concerns remain concerning my original comments.

1. The chromosome counting (aneuploidy) experiment is an excellent addition. However, whereas the control is clear, the cVCA example shown appears incorrect. #5 is in fact two univalents. #4 and #12 are probably individual chromatids, and #7/17 are not resolved enough to count (at least in this z-projection). Please pick better examples, and show individual slices to convince the reader if necessary.
2. Fig 2B and C is a good addition, and a reasonable alternative to the experiment requested. Since it remains the case that sirtubulin could be turning over/exchanging in the cytoplasm, plus previously stated concerns about MT types in oocytes, the negative result is as such meaningless and it is misleading to leave this in the manuscript. Ie fig 2E should be removed.

Response to Reviewers' s comments:

Reviewer #1 (Remarks to the Author):

The authors made major attempts to address the criticism raised by me and the other reviewers. These clarified important points and significantly strengthened the conclusions. Overall, I would be happy to support the publication of this revised version in Nature Communications. However, I would have two additional remarks:

1. While the authors convincingly show in their rebuttal letter that they have made any reasonable attempt to more directly address the role of myosin II, unfortunately all these attempts failed. Overall, the findings are important and are now supported by various lines of indirect evidence, for which reason I am in support of publication. However, the discussion should still be formulated carefully to point out the lack of direct evidence in support of this main conclusion.

We agree with reviewer#1 comments. Following his/her advice, we also changed our title to: "Artificially decreasing cortical tension generates aneuploidy in mouse oocytes".

2. Especially for the revised sections, the phrasing is rather imprecise at certain places. For example, by saying "This tedious computational approach blindly quantified..." they probably mean that the approach is rigorous and is unbiased. Similarly, at a later point, they appear to draw a conclusion from statistically non-significant differences. They should provide an explanation for this. Maybe the statistical test was not appropriate?

We agree with the reviewer and modified these sections accordingly.

Reviewer #2 (Remarks to the Author):

I am impressed by how well the authors have addressed my concerns. By adding in additional studies with cFH1FH2 they convincingly show similar results with cVCA and cFH1FH2, most crucially now showing increased cytoplasmic myosin II with cFH1FH2 just as with cVCA (FigS5). I also appreciate that my suggestion to investigate naturally occurring soft oocytes was a big ask but I am happy that authors went some way to address this by reducing myosin II activity in control embryos and finding a slight improvement in chromosome alignment (Fig5). I am therefore happy to recommend publication of the revised manuscript in Nature Communications.

No request.

Reviewer #3 (Remarks to the Author):

The authors have made considerable effort to improve the manuscript in line with reviewers comments. The following two concerns remain concerning my original comments.

1. The chromosome counting (aneuploidy) experiment is an excellent addition. However, whereas the control is clear, the cVCA example shown appears incorrect. #5 is in fact two univalents. #4 and #12 are probably individual chromatids, and #7/17 are not resolved enough to count (at least in this z-projection). Please pick better examples, and show individual slices to convince the reader if necessary.

We agree with the referee that the cVCA example is less clear than the control one if we look only at the projected images (supplementary Figure 1). It was the same in the original paper describing this technique and suggested by the reviewer (Figure 3 of Duncan et al 2009). However, the chromosome count is clear if one watches the Supplementary movie 1 showing all the individual slices.

2. Fig 2B and C is a good addition, and a reasonable alternative to the experiment requested. Since it remains the case that sirtubulin could be turning over/exchanging in the cytoplasm, plus previously stated concerns about MT types in oocytes, the negative result is as such meaningless and it is misleading to leave this in the manuscript. Ie fig 2E should be removed.

We agree with the referee and removed Figure 2E.